# Enhanced Causal Discovery for Autocorrelated Time Series via Adaptive Momentary Conditional Independence

## Abstract

Discovering causal relationships from time series data is essential for understanding complex dynamical systems across a range of domains. However, strong autocorrelation often limits the detection power of existing algorithms and increases the risk of false positives. Moreover, when both lagged and contemporaneous links are considered, existing algorithms are prone to generating false positives in lagged link detection due to indirect causal effects induced by contemporaneous mediators. To address these challenges, the Adaptive Momentary Conditional Independence (aMCI) method is introduced to mitigate the masking effects of autocorrelation and maintain control over false discovery rates. The aMCI adaptively modifies the conditioning set while aggregating conclusions from multiple conditional independence tests. In addition, a multi-phase algorithm is proposed to robustly learn the causal graph by effectively applying the aMCI. The algorithm is designed to be hyperparameter-insensitive, order-independent, and provably consistent under oracle conditions. Extensive evaluations on simulated and benchmark datasets demonstrate that the proposed algorithm substantially improves the accuracy of causal discovery from time series, especially in detecting lagged links.

## 1 Introduction

Multivariate time series exist widely in various domains such as earth science, neuroscience, and economics. Discovering causal relationships within these multivariate time series—encompassing both contemporaneous and lagged effects—is crucial for understanding the underlying mechanisms driving these systems. Accurate causal structures enable researchers to better comprehend system dynamics, build more precise prediction models, and estimate causal effects more reliably within the potential outcome framework.

While significant advances have been made in causal discovery methods for time series data, a persistent challenge remains: effectively handling autocorrelation (Nogueira et al., 2022; Gong et al., 2024). Autocorrelation, the correlation of a variable with its past values, is a common characteristic of time series data that can significantly impact the performance of causal discovery algorithms. In particular, strong autocorrelation can obscure true causal relationships, leading to both false negatives (missed causal links) and false positives (spurious causal links) (Runge et al., 2019).

Existing approaches for causal discovery from time series data broadly fall into four categories: constraint-based, score-based, structural causal model-based, and Granger causality-based algorithms. Optimization-based algorithms like NOTEARS (Zheng et al., 2018) and its nonlinear extension NTS-NOTEARS (Sun et al., 2023) formulate directed acyclic graph (DAG) learning as continuous optimization problems. These methods have shown promising results but may require careful hyperparameter tuning, which can present challenges in some practical applications. Alternatively, constraint-based algorithms such as PCMCI (Runge et al., 2019) and PCMCI+ (Runge, 2020) utilize conditional independence tests to learn causal graphs from time series. PCMCI+ incorporates contemporaneous links discovery while effectively controlling false positive rates. However, its detection power for lagged links tends to decrease as autocorrelation increases. Bagged-PCMCI+ (Debeire et al., 2024) addresses uncertainty estimation through bagging techniques, though at the expense of greatly increased computational requirements. Structural causal model-based and Granger

causality-based algorithms are not discussed in detail here due to differences in their causal graph outputs, with comprehensive explanations provided in the Appendix A.2.

To address these challenges, a novel method called Adaptive Momentary Conditional Independence (aMCI) is proposed. Conditional independence testing forms the backbone of decision-making in the constraint-based causal discovery algorithms, and thus enhancing its efficiency can substantially influence overall performance. The aMCI achieves this goal by dynamically modifying the conditioning set according to the causal structure of variables. Different strategies are applied to lagged and contemporaneous links, with a structured decision procedure guiding the choice of an appropriate conditioning set for each conditional independence test. Building on this foundation, the paper introduces the Enhanced Causal Discovery via aMCI (ECD-aMCI) algorithm specifically designed to fully leverage the capabilities of aMCI. Conventional causal discovery algorithms typically start from a fully connected graph, which can limit aMCI's effectiveness in the initial stages. To overcome this limitation, ECD-aMCI employs a progressive refinement strategy: first establishing initial lagged parent estimates, then refining these estimates using aMCI, before finally discovering the complete causal skeleton. This sequential approach allows aMCI to operate with increasingly accurate structural information, substantially improving detection power for true causal links in autocorrelated systems while maintaining robust control over false positive rates.

The contributions of this paper are summarized as follows.

- The development of aMCI, a new method that strategically modifies conditioning sets in time series data to overcome the masking effects of autocorrelation.
- A multi-phase algorithm ECD-aMCI is proposed to fully leverages the capabilities of aMCI through progressive refinement of causal structures. The proposed algorithm provides a hyperparameter-insensitive, order-independent, proven consistency framework to learn both lagged and contemporaneous links.
- This paper offers a novel perspective, emphasizing that for constraint-based algorithms, theoretically equivalent design choices under ideal conditions may yield different results in practice, necessitating a preference for choices better suited to the data.

The remainder of this paper is organized as follows. Section 2 introduces the fundamental concepts of d-separation in causal graphs and the PCMCI algorithm. Section 3 describes the aMCI method and the ECD-aMCI algorithm in detail. Section 4 provides a comprehensive evaluation of the ECD-aMCI algorithm through both simulated and benchmark datasets. Finally, Section 5 concludes with a summary of our contributions. The code implementation of the proposed algorithm is available in the Supplementary Material.

## 2 PRELIMINARIES

### 2.1 D-SEPARATION AND COMPLETED PARTIALLY DIRECTED ACYCLIC GRAPH

A directed acyclic graph (DAG) (Pearl, 2009) encodes a set of conditional independence relations through the criterion of *d-separation*. Given three disjoint subsets of nodes $X$, $Y$, and $Z$ in a DAG $\mathcal{G}$, we say that $X$ is d-separated from $Y$ given $Z$ (denoted $X \perp\!\!\!\perp Y \mid Z$) if all paths from any node in $X$ to any node in $Y$ are blocked by $Z$ according to the following rules: (1) A non-collider path segment $A \to B \to C$, $A \leftarrow B \to C$, or $A \leftarrow B \leftarrow C$ is blocked if the middle node $B \in Z$; (2) A collider path segment $A \to B \leftarrow C$ is blocked unless $B \in Z$ or some descendant of $B$ is in $Z$. D-separation provides a graphical criterion to determine whether two variables are conditionally independent given a set of other variables. It forms the theoretical basis for constraint-based causal discovery algorithms.

Completed Partially Directed Acyclic Graph (CPDAG) (Spirtes et al., 2002) represents a Markov equivalence class of DAGs by displaying directed edges where causal direction is consistent across all equivalent graphs, and undirected edges where direction cannot be uniquely determined from observational data alone.

### 2.2 THE PCMCI ALGORITHM

PCMCI is a constraint-based causal discovery algorithm specifically designed for high-dimensional time series data. It extends the PC algorithm by addressing temporal dependencies and controlling

for autocorrelation. The PCMCI algorithm consists of two main steps: (1) PC$_1$ step: A preliminary skeleton is constructed by iteratively removing links based on a set of conditional independence tests, using a gradually increasing conditioning set. (2) MCI step: The resulting links are then subjected to a stricter conditional independence test, called the Multivariate Conditional Independence (MCI) test, which conditions on both parents of the target and source variables to better control false positives.

The Multivariate Conditional Independence (MCI) test is designed to evaluate the conditional independence between two time series variables $X_{i,t-\tau}$ and $X_{j,t}$, while accounting for temporal dependencies. Specifically, it tests

$$X_{i,t-\tau} \perp\!\!\!\perp X_{j,t} \mid \mathcal{P}(X_{i,t-\tau}) \cup \mathcal{P}(X_{j,t}),$$

where $\mathcal{P}(\cdot)$ denotes the estimated parent set of a variable. This choice of conditioning set, including both source and target parents, helps to mitigate the effects of autocorrelation and confounding from nearby variables.

## 3 METHOD

This section describes the proposed causal discovery algorithm for time series through three parts. First, Section 3.1 illustrates the intuitive insights of aMCI. Section 3.2 introduces the Adaptive Momentary Conditional Independence (aMCI) method to improve detection power by refining the conditioning sets used in tests. Finally, Section 3.3 presents a multi-phase algorithm to effectively leverage the aMCI method. Theoretical properties for the algorithm are provided in Section 3.4.

### 3.1 INTUITIVE INSIGHTS OF aMCI

For intuitive understanding of aMCI, consider a bivariate time series $\{X_t\}_{t\in\mathbb{N}_+}$, where $X_t = (X_{1,t}, X_{2,t})'$. This time series is generated according to the following structural causal model:

$$
\begin{aligned}
X_{1,t} &= aX_{1,t-1} + \varepsilon_{1,t}, \\
X_{2,t} &= aX_{2,t-1} + cX_{1,t-1} + \varepsilon_{2,t},
\end{aligned}
\tag{1}
$$

where $a$ represents the strength of autocorrelation, $c$ represents the strength of the causal link $X_{1,t-1} \to X_{2,t}$, and $\varepsilon_{1,t}, \varepsilon_{2,t}$ are independent noise terms. The corresponding full causal graph is depicted in Figure 1A.

According to the MCI defined in Section 2, determining whether the link $X_{1,t-1} \to X_{2,t}$ exists involves testing whether $X_{1,t-1} \perp\!\!\!\perp X_{2,t} \mid \mathcal{B}^-(X_{1,t-1}) \cup \mathcal{B}^-(X_{2,t})$ holds, where $\mathcal{B}^-(X_{1,t-1}) = \{X_{1,t-2}\}$ and $\mathcal{B}^-(X_{2,t}) = \{X_{2,t-1}\}$ in model (1). Although from the d-separation perspective, conditioning on $X_{1,t-2}$ should not affect the conditional dependence between $X_{1,t-1}$ and $X_{2,t}$, simulations reveal that replacing $\mathcal{B}^-(X_{1,t-1})$ with $\mathcal{B}_{ad}^-(X_{1,t-1}) = \{X_{1,t-3}\}$ (i.e., substituting $X_{1,t-2}$ with $X_{1,t-3}$) significantly improves the detection power for the link $X_{1,t-1} \to X_{2,t}$.

This phenomenon can be explained through equations (2)-(3). Equations (2) and (3) can be derived straightforwardly from model (1). The distinction between equations (2) and (3) lies in the substitution of $X_{1,t-2}$ in equations (2) with $X_{1,t-3}$ and $\varepsilon_{1,t-2}$.

$$
\begin{aligned}
X_{1,t-1} &= \underbrace{aX_{1,t-2}}_{\text{conditioning}} + \underbrace{\varepsilon_{1,t-1}}_{\text{randomness}}, \\
X_{2,t} &= \underbrace{aX_{2,t-1} + caX_{1,t-2}}_{\text{conditioning}} + \underbrace{c\varepsilon_{1,t-1} + \varepsilon_{2,t}}_{\text{randomness}},
\end{aligned}
\tag{2}
$$

$$
\begin{aligned}
X_{1,t-1} &= \underbrace{a^2X_{1,t-3}}_{\text{conditioning}} + \underbrace{a\varepsilon_{1,t-2} + \varepsilon_{1,t-1}}_{\text{randomness}}, \\
X_{2,t} &= \underbrace{aX_{2,t-1} + ca^2X_{1,t-3}}_{\text{conditioning}} + \underbrace{ca\varepsilon_{1,t-2} + c\varepsilon_{1,t-1} + \varepsilon_{2,t}}_{\text{randomness}},
\end{aligned}
\tag{3}
$$

In equations (2), the randomness of $X_{1,t-1} \mid X_{1,t-2}, X_{2,t-1}$ depends on the error $\varepsilon_{1,t-1}$, while the randomness of $X_{2,t} \mid X_{1,t-2}, X_{2,t-1}$ depends on $c\varepsilon_{1,t-1} + \varepsilon_{2,t}$. Thus, the conditional independence test $X_{1,t-1} \perp\!\!\!\perp X_{2,t} \mid X_{1,t-2}, X_{2,t-1}$ relies on detecting the correlation between $\varepsilon_{1,t-1}$ and

$c\varepsilon_{1,t-1} + \varepsilon_{2,t}$. In contrast, in equations (3), the randomness of $X_{1,t-1} \mid X_{1,t-3}, X_{2,t-1}$ depends on $a\varepsilon_{1,t-2} + \varepsilon_{1,t-1}$, while the randomness of $X_{2,t} \mid X_{1,t-3}, X_{2,t-1}$ depends on $ca\varepsilon_{1,t-2} + c\varepsilon_{1,t-1} + \varepsilon_{2,t}$. Therefore, the conditional independence test $X_{1,t-1} \perp\!\!\!\perp X_{2,t}|X_{1,t-3}, X_{2,t-1}$ relies on detecting the correlation between $a\varepsilon_{1,t-2} + \varepsilon_{1,t-1}$ and $ca\varepsilon_{1,t-2} + c\varepsilon_{1,t-1} + \varepsilon_{2,t}$. Notably, the randomness in Equations (3) both contain the common components $\varepsilon_{1,t-2}$ and $\varepsilon_{1,t-1}$ with amplifying coefficients. This structure produces a stronger correlation than the single shared term $\varepsilon_{1,t-1}$ in Equations (2), thereby enhancing the statistical power for detecting the causal link. This analytical process is visually illustrated in Figures 1C and 1D.

However, this substitution is not always beneficial. In Figure 1B, $X_{1,t-2}$ acts as a confounder for both $X_{1,t-1}$ and $X_{2,t}$. Failing to condition on this confounder could lead to false positive detection of the link $X_{1,t-1} \to X_{2,t}$. This highlights the need for an adaptive method that adjusts the conditioning set based on the specific causal structure, such as conditioning on $X_{1,t-2}$ when it is a confounder (Figure 1B) but conditioning on an alternative variable (e.g., $X_{1,t-3}$) when it is not (Figure 1A).

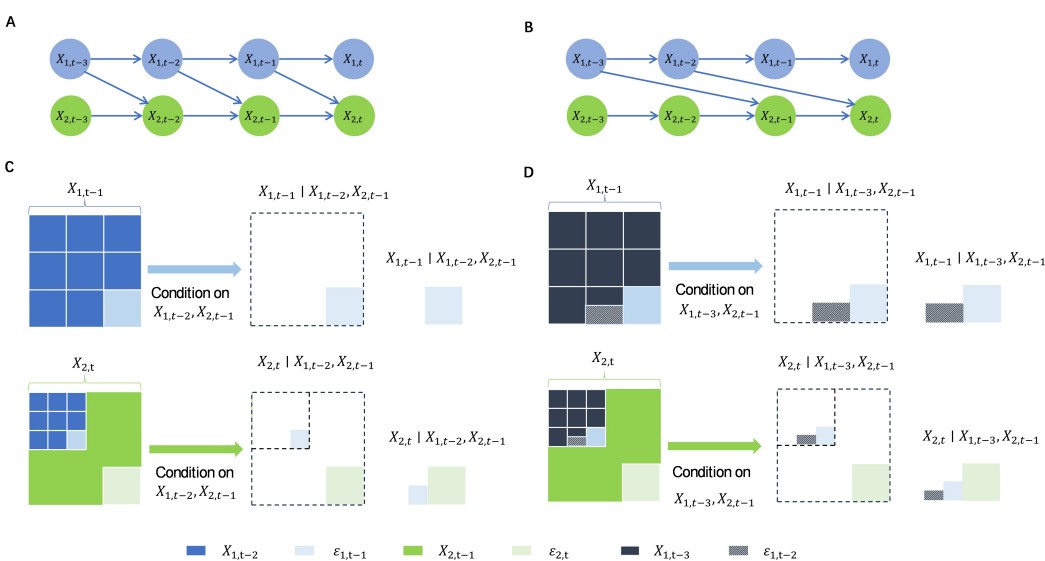

Figure 1: **(A)** Causal graph corresponding to the model (1). **(B)** Alternative causal graph with $X_{1,t-2}$ as a confounder. **(C)** Causal signal loss when conditioning on $X_{1,t-2}$ and $X_{2,t-1}$ under causal graph **A**. **(D)** Causal signal enhancement when conditioning on $X_{1,t-3}$ and $X_{2,t-1}$ under causal graph **A**.

## 3.2 Adaptive Momentary Conditional Independence

As established in Section 3.1, original methods such as MCI, while effective in controlling false positives, may exhibit reduced detection power when applied to time series with strong autocorrelation. The core issue identified is that conditioning on highly correlated immediate predecessors of source variable $X_{i,t-1}$ (e.g., $X_{i,t-2}$ when testing $X_{i,t-1} \perp\!\!\!\perp X_{j,t} \mid X_{i,t-2}$) can obscure the signal of the direct link, even though it does not block the path according to d-separation. To address this challenge, this paper proposes the aMCI method that dynamically modifies the conditioning set based on the temporal structure of the variables in conditioning set. The key innovation of aMCI lies in its strategic handling of immediate predecessors—replacing them with earlier variables when they might obscure causal signals rather than control for confounding.

Using the time series data $\boldsymbol{X}$ and a chosen conditional independence test, the operation of aMCI can be formally represented as the mapping:

$$\text{aMCI} : (X_{i,t-\tau}, X_{j,t}, \mathcal{S}, \hat{\mathcal{B}}^-(X_{i,t-\tau}), \hat{\mathcal{B}}^-(X_{j,t})) \to (p\text{-value}, I, \mathcal{S}_{ad}),$$

where $X_{i,t-\tau}$ and $X_{j,t}$ are the specific variables under investigation for conditional independence (representing a potential cause and effect, respectively, with lag $\tau \geq 0$); $\mathcal{S}$ is the initial set of conditioning variables provided to aMCI; $\hat{\mathcal{B}}^-(\cdot)$ represents the estimated set of lagged parents for a given variable; $\mathcal{S}_{ad}$ is the final, adaptively determined, conditioning set generated by the aMCI

based on $\mathcal{S}$; and $I$ is the test statistic value associated with this $p$-value. The aMCI handles different scenarios based on the temporal relationship between $X_{i,t-\tau}$ and $X_{j,t}$.

**Case 1 (Lagged Links ($\tau > 0$))** *The aMCI checks whether $X_{i,t-\tau-k}$ appears in both parent sets $\hat{\mathcal{B}}^-(X_{i,t-\tau})$ and $\hat{\mathcal{B}}^-(X_{j,t})$, where $X_{i,t-\tau-k} \in \hat{\mathcal{B}}^-(X_{i,t-\tau})$ and $k \in \{1,\ldots,\tau_{max}\}$. If so, the original conditioning set is maintained. Otherwise, $X_{i,t-\tau-k}$ is substituted with $X_{i,t-\tau-k-1}$ in the conditioning set to avoid obscuring causal signals.*

A signal-to-noise ratio (SNR)-based explanation is provided for why testing $X_{i,t-\tau} \perp\!\!\!\perp X_{j,t} \mid X_{i,t-\tau-k-1}$ can be preferable to testing $X_{i,t-\tau} \perp\!\!\!\perp X_{j,t} \mid X_{i,t-\tau-k}$ when $X_{i,t-\tau-k}$ is not a confounder and the true causal structure is:

$$X_{i,t-\tau-k} \to X_{i,t-\tau} \to X_{j,t}.$$

Let the information contained in $X_{j,t}$ be denoted as $I(X_{j,t})$, and the information flow from $X_{i,t-\tau}$ to $X_{j,t}$ be denoted as $I(X_{i,t-\tau} \to X_{j,t})$. In testing for independence between $X_{i,t-\tau}$ and $X_{j,t}$, $I(X_{i,t-\tau} \to X_{j,t})$ represents the useful signal, while the remaining information in $I(X_{j,t})$ is treated as noise. A stronger signal implies a more detectable causal effect and reduces the risk of false negatives.

When conditioning on $X_{i,t-\tau-k}$, the signal in $X_{j,t}$ is correspondingly reduced and becomes $I(X_{i,t-\tau} \to X_{j,t}) - I(X_{i,t-\tau-k} \to X_{i,t-\tau} \to X_{j,t})$. The corresponding SNR is:

$$SNR_{X_{j,t}} = \frac{I(X_{i,t-\tau} \to X_{j,t}) - I(X_{i,t-\tau-k} \to X_{i,t-\tau} \to X_{j,t})}{I(X_{j,t}) - [I(X_{i,t-\tau} \to X_{j,t}) - I(X_{i,t-\tau-k} \to X_{i,t-\tau} \to X_{j,t})]}.$$

From basic mathematical analysis, $SNR_{X_{j,t}}$ is a decreasing function of $I(X_{i,t-\tau-k} \to X_{i,t-\tau} \to X_{j,t})$. When replacing $X_{i,t-\tau-k}$ with $X_{i,t-\tau-k-1}$, we have $I(X_{i,t-\tau-k-1} \to X_{i,t-\tau} \to X_{j,t}) \leq I(X_{i,t-\tau-k} \to X_{i,t-\tau} \to X_{j,t})$, assuming $X_{i,t-\tau-k-1}$ transmits less information through the causal chain (Assumption A4 in Section 3.4). This reduction thus improves $SNR_{X_{j,t}}$. As a result, the conditioning set used in aMCI is more likely to lead to correct decisions and fewer false negatives.

**Case 2 (Contemporaneous Links ($\tau = 0$))** *The aMCI employs a more nuanced strategy depending on whether $X_{i,t-k}$ and $X_{j,t-k}$ are confounders of $(X_{i,t}, X_{j,t})$, where $X_{i,t-k} \in \hat{\mathcal{B}}^-(X_{i,t})$ or $X_{j,t-k} \in \hat{\mathcal{B}}^-(X_{j,t})$, $k \in \{1,\ldots,\tau_{max}\}$. (1) If both are confounders, the standard conditioning set is maintained. (2) If only one is a confounder, the non-confounder immediate predecessor is replaced with its earlier lag. (3) If neither is a confounder, three possible conditioning sets are evaluated (substituting either immediate predecessor or using the standard set) and the one yielding the smallest p-value is selected.*

It is important to observe that the handling of lagged links follows directly from the intuition provided in Section 3.1. However, the handling of contemporaneous links requires a more sophisticated approach. This complexity arises because the aMCI methodology involves adjusting the conditioning set of the source variable, yet for contemporaneous links $(X_{i,t}, X_{j,t})$, temporal precedence cannot be used to determine which variable serves as the source. Consequently, **Case 2** explores three scenarios based on the principle that confounders must be retained in the conditioning set. Specifically, in **Case 2** (3), by considering test results from three different conditioning sets and selecting the one that yields the smallest p-value, the method ensures that links previously undetectable due to autocorrelation become readily identifiable. The pseudocode of aMCI is provided in Algorithm 1 of Appendix A.3.

The primary challenge in implementing such an adaptive method arises because the true causal structure, particularly the role of immediate predecessors, is typically unknown and must be inferred. Conventional constraint-based discovery algorithms learn the structure iteratively, typically beginning with a fully connected graph structure. It is difficult to apply such adaptive logic effectively in the early stages when the necessary structural information is still uncertain. This challenge motivates the multi-phase algorithm detailed in Section 3.3.

## 3.3 Enhanced Causal Discovery Algorithm

Traditional causal discovery algorithms typically start from a fully connected graph and progressively remove edges through conditional independence tests. However, the traditional procedure limits the

effectiveness of aMCI in the initial stages, as in a fully connected graph, immediate predecessors are necessarily treated as confounders. To fully leverage the capabilities of aMCI, this paper proposes the ECD-aMCI algorithm that operates in three sequential phases.

The core innovation of the ECD-aMCI algorithm lies in its progressive refinement approach, which enables more accurate estimation of causal structures in time series data with autocorrelation. The algorithm consists of three phases: (1) initial estimation of lagged parent sets, (2) refinement of these estimated sets using aMCI, and (3) discovery of the complete causal structure including both lagged and contemporaneous links.

**Phase 1: PC$_1$-Based Initial Estimation.** In the first phase, a simplified PC algorithm is employed to obtain initial estimates of the lagged parent sets. This phase considers only lagged relationships without accounting for contemporaneous effects, which may lead to false positives due to indirect causal effects mediated by contemporaneous variables. Starting with a fully connected lagged structure (up to maximum lag $\tau_{\max}$), the algorithm iteratively tests the conditional independence between each lagged variable $X_{i,t-\tau}$ and target variable $X_{j,t}$, given progressively larger conditioning sets $\mathcal{S}$. The minimum test statistic values are stored and links that are conditionally independent given any conditioning set are removed. The procedure continues until all possible conditioning sets up to the available size (up to one) have been tested. The result is an initial estimate of lagged parent sets $\hat{\mathcal{B}}^-(X_{j,t})$ for each variable $X_{j,t}$.

**Phase 2: Lagged Parent Sets Refinement via aMCI.** Since the initial estimates may include false positives due to contemporaneous mediator, the second phase refines these estimates using aMCI. Starting with the initial estimates from Phase 1, the algorithm tests the conditional independence of adjacent pairs $(X_{i,t-\tau}, X_{j,t})$ for $\tau > 0$ using aMCI. The conditioning sets $\mathcal{S}$ are chosen from contemporaneous adjacency set $\hat{\mathcal{A}}(X_{j,t})$, and the algorithm progressively increases the size of these conditioning sets up to one. After each iteration, the lagged parent sets are updated based on the current graph structure, which enables more accurate aMCI tests in subsequent iterations.

**Phase 3: Complete Skeleton Discovery.** The final phase builds on the refined lagged parent sets of Phase 2 to discover the complete causal skeleton, including both lagged and contemporaneous relationships. This phase ensures robust results regardless of the variable order by not immediately updating $\hat{\mathcal{B}}(X_{j,t})$ and $\hat{\mathcal{A}}(X_{j,t})$ based on aMCI test results. Unlike conventional constraint-based algorithms, this phase tests conditional independence for lagged links first and then contemporaneous links, which minimizes the influence of not-yet-removed edges on the effectiveness of aMCI. For lagged relationships, the algorithm tests and potentially removes links $X_{i,t-\tau} \to X_{j,t}$. For contemporaneous relationships, it tests and potentially removes links $X_{i,t} \leftrightarrow X_{j,t}$. After each iteration, both the lagged parent sets and contemporaneous adjacency sets are updated based on the current graph structure.

By progressively refining estimates and leveraging the capabilities of aMCI, the three phases outlined above enable more accurate identification of causal adjacencies in time series. The subsequent orientation rules in the ECD-aMCI algorithm follow the Meek's rules (Meek, 1995) and build upon the relevant work of Runge (2020). The complete pseudocode of the ECD-aMCI algorithm is provided in the Appendix A.3.

### 3.4 THEORETICAL PROPERTIES

This section provides a theoretical analysis of the ECD-aMCI algorithm. First, necessary assumptions for the analysis are introduced. Then, three key theoretical properties of ECD-aMCI are established: consistency, order independence and enhanced accuracy. Detailed proofs of all theorems are provided in the Appendix A.4.2.

**Assumptions** The theoretical analysis relies on several standard assumptions in time series causal discovery: (A1) Causal Sufficiency, (A2) Causal Markov Condition, (A3) Faithfulness, (A4) Temporal Priority, and (A5) Stationarity. These assumptions are commonly adopted in the causal discovery literature (Spirtes et al., 2001; Runge et al., 2019; Runge, 2020; Biswas & Shlizerman, 2022). Detailed definitions of these assumptions are provided in the Appendix A.4.1.

**Theorem 1 (Consistency)** *Under the Assumptions (A1)-(A5), if the conditional independence tests are oracle, ECD-aMCI returns the correct CPDAG, i.e., $\hat{\mathcal{G}} = \mathcal{G}_{CPDAG}$, where $\mathcal{G}_{CPDAG}$ denotes the CPDAG of the time series graph $\mathcal{G}$.*

Theorem 1 demonstrates that given oracle conditional independence information, the ECD-aMCI algorithm guarantees correct identification of all links that can be identified based on conditional independence information. The proof strategy involves first establishing that the proposed algorithm correctly identifies the skeleton of the graph, and then verifying that the orientation rules correctly identify directions. This establishes that ECD-aMCI achieves the same theoretical consistency guarantees as other constraint-based causal discovery algorithms under ideal conditions.

**Theorem 2 (Enhanced Accuracy)** *Under the Assumptions (A1)-(A5), applying the aMCI method yields a higher probability of learning the correct causal graph, i.e., $P(\hat{\mathcal{G}} = \mathcal{G}_{CPDAG}) \geq P(\tilde{\mathcal{G}} = \mathcal{G}_{CPDAG})$, where $\tilde{\mathcal{G}}$ represents the causal graph estimated without applying the aMCI method.*

Theorem 2 shows that the aMCI method improves the accuracy of causal graph recovery in autocorrelated time series. The combination of Theorems 1 and 2 establishes that the ECD-aMCI algorithm offers a novel perspective on selecting more powerful designs of constraint-based algorithms among those that are theoretically equivalent under ideal conditions but exhibit different practical performance. This perspective has the potential to extend to broader scenarios beyond time series.

Since Colombo & Maathuis (2014) proposed the PC-stable algorithm, order independence has been recognized as an essential property for causal discovery algorithms. The proposed algorithm avoids the influence of variable ordering in all edge removal processes based on conditional independence tests and in the orientation phase, naturally leading to Theorem 3.

**Theorem 3 (Order Independence)** *Under the Assumptions (A1)-(A5), the outcome of ECD-aMCI is independent of the order of the variables.*

## 4 EVALUATION

This section presents a comprehensive evaluation of the ECD-aMCI algorithm. First, several baseline algorithms for comparison are introduced, followed by the evaluation metrics used to assess performance. The data generation process for simulations is then described in detail. Results from both simulated and benchmark datasets are analyzed to demonstrate the effectiveness of ECD-aMCI. Finally, hyperparameter settings for all algorithms are discussed.

**Baselines** ECD-aMCI is compared with several recent causal discovery algorithms: PCMCI+, Bagged-PCMCI+, and NTS-NOTEARS. The constraint-based methods (PCMCI+, Bagged-PCMCI+, and ECD-aMCI) can be directly compared as they are all implemented with ParCorr (Hotelling, 1953) test for linear cases and GPDC (Runge, 2018) test for nonlinear cases. In contrast, NTS-NOTEARS is a continuous optimization-based algorithm whose hyperparameter weight threshold $W_{thresh}$ does not have common choices like the confidence level $\alpha$ in constraint-based algorithms (e.g., $\alpha = 0.01$). The weight threshold $W_{thresh}$ is adjusted through grid search to ensure that its $F_1$-score is maximized.

**Evaluation Metrics** To thoroughly assess the performance of causal discovery methods, four important metrics are employed that evaluate different aspects of algorithms. The $F_1$-score, which balances precision and recall, is measured seperately for lagged cross-adjacencies ($i \neq j$) and all adjacencies, with higher values indicating better performance. The Structural Hamming Distance (SHD) is measured to quantify the overall structural difference between the learned and true causal graphs. SHD counts the number of edge additions, deletions, and reversals needed to transform one graph into another, with lower values indicating better performance. The average runtimes were evaluated on Intel Xeon Platinum 8260L CPU at 2.30GHz. To remove the influence of patterns in marginal variance that might affect results, all metrics were computed on standardized data (Reisach et al., 2021). Additional metrics including $F_1$-scores for contemporaneous and autodependency adjacencies are provided in the Appendix A.5.

**Simulated Data Generation** Following Runge et al. (2019) and Runge (2020), autocorrelated time series with both contemporaneous and lagged causal dependencies are generated. The data is

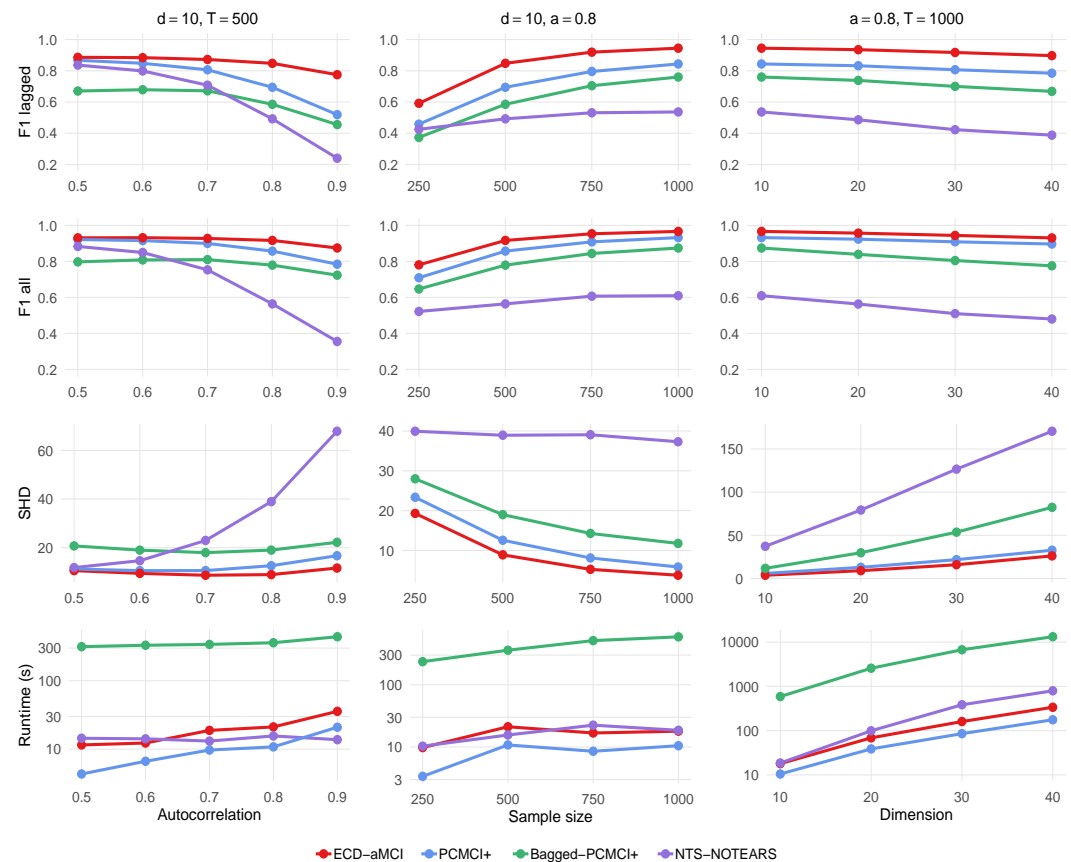

Figure 2: Mean metrics over 300 datasets for each linear setting.

generated from an additive model where each variable is influenced by its own past value, the past and contemporaneous values of its causal parents, and random noise. Specifically, the following model is used:

$$X_{j,t} = \sum_{\tau=1}^{\tau_{\max}} \delta_{j,j,\tau} a_j f_j(X_{j,t-\tau}) + \sum_{i=1}^{d} \sum_{\tau=1}^{\tau_{\max}} \delta_{i,j,\tau} c_{i,j,\tau} f_i(X_{i,t-\tau}) + \varepsilon_{j,t}, \qquad (4)$$

where $X_{j,t}$ represents the value of variable $j$ at time $t$, $j \in \{1, \ldots, d\}$, $d$ is the dimension of time series. The autocorrelation coefficients $a_j$ are uniformly drawn from $[\max(0, a - 0.2), a]$, where $a$ controls the autocorrelation strength. The noise term $\varepsilon_{j,t}$ follows an i.i.d. standard Gaussian distribution. The coefficient $c_{i,j,\tau}$, which represents the strength of the link $X_{i,t-\tau} \to X_{j,t}$, is drawn uniformly from $[0.15, 0.25]$ to ensure the stationarity of the time series. The coefficient $\delta_{i,j,\tau}$ is a binary variable indicating the existence of the link $X_{i,t-\tau} \to X_{j,t}$. For each model, $2.5 \cdot d$ cross-links between variables are randomly selected. 40% of the links are set to be contemporaneous ($\tau = 0$), and the remaining links have time lags $\tau$ uniformly drawn from $\{1, \ldots, \tau_{\text{true}}\}$, where true maximum time lag $\tau_{\text{true}}$ is set to 5. The functional dependencies $f_{i,j}(x)$ are either linear functions or nonlinear functions $\tanh(x)$. Each simulated dataset is generated with different randomly sampled parameters and a different ground truth causal graph. Only stationary models are considered to ensure the validity of the causal discovery task.

**Results** The simulation experiments systematically evaluate performance across three key parameters: autocorrelation strength ($a$), sample size ($T$), and dimension of the time series ($d$). As shown in Fig.2, for each parameter setting, ECD-aMCI consistently outperforms the baseline algorithms across all evaluation metrics. In the first column, as the autocorrelation strength increases from 0.5 to 0.9, ECD-aMCI maintains superior F1-scores for lagged and all adjacencies while keeping the SHD lower than competing methods. The advantage becomes more pronounced as autocorrelation strengthens. The second column demonstrates that ECD-aMCI achieves better performance even with

limited sample sizes. The third column illustrates that ECD-aMCI scales effectively with increasing dimensionality, maintaining high F1-scores and low SHD as the dimension increases from 10 to 40. Regarding computational efficiency, the runtime comparison in the bottom row shows that ECD-aMCI achieves this superior performance with reasonable computational cost. These metrics in Figure 2 are computed based on 300 independently generated datasets. Detailed numerical results including means and standard deviations, as well as results from nonlinear function settings, are provided in the Appendix A.5.2.

**Benchmark Data**  The functional Magnetic Resonance Imaging (fMRI) benchmark from NetSim contains rich, realistic simulated time-series for modeling brain networks (Smith et al., 2011). This nonlinear benchmark dataset has been widely used to evaluate time series causal discovery algorithms due to its diverse range of underlying networks that closely mimic challenges in neuroimaging analysis. As demonstrated in Table 1, ECD-aMCI achieves higher F1-scores and lower SHD on this dataset compared to all baseline methods, confirming its effectiveness and practical value for neuroscientific applications. Due to the prohibitively high computational cost of the Bagged-PCMCI+ algorithm when processing non-linear datasets, this algorithm is excluded from consideration in the analysis of fMRI datasets.

Table 1: Performance comparison on 50 fMRI datasets with 200 observations each.

| Method | $d = 5$ | | $d = 10$ | |
|---|---|---|---|---|
| | F1-score$_{\text{all}}$ | SHD | F1-score$_{\text{all}}$ | SHD |
| ECD-aMCI | $0.908 \pm 0.075$ | $5.940 \pm 1.737$ | $0.843 \pm 0.048$ | $17.840 \pm 2.129$ |
| NTS-NOTEARS | $0.672 \pm 0.056$ | $11.040 \pm 2.218$ | $0.646 \pm 0.039$ | $28.060 \pm 3.706$ |
| PCMCI+ | $0.871 \pm 0.081$ | $6.560 \pm 1.651$ | $0.809 \pm 0.059$ | $19.280 \pm 2.307$ |

**Hyperparameters**  ECD-aMCI has two hyperparameters. The first is the confidence level $\alpha$ for hypothesis testing, which should be set as low as possible while maintaining sufficient detection power. In all experiments presented in this paper, $\alpha = 0.01$ is used for all constraint-based methods (ECD-aMCI, PCMCI+, and Bagged-PCMCI+). The second hyperparameter is the maximum time lag $\tau_{\max}$, which in all experiments is set to the true maximum time lag. In practical applications, $\tau_{\max}$ can be initially chosen based on estimated autocorrelation coefficients, typically selecting a slightly larger value, and then iteratively reduced based on the estimated causal graph. Simulation results in Table 2 indicate that the proposed method exhibits robustness to the choice of $\tau_{\max}$. NTS-NOTEARS has six hyperparameters, with the weight threshold $W_{\text{thresh}}$ being sensitive to the strength of causal links. Therefore, the $W_{\text{thresh}}$ is determined by maximizing the F1-score, while other hyperparameters are set to the optimal values reported in the original paper (Sun et al., 2023). See the Appendix A.5 for detailed hyperparameter settings for all methods.

Table 2: Robustness of the ECD-aMCI to hyperparameter $\tau_{\max}$.

| | $\tau_{\max} = 5$ | $\tau_{\max} = 6$ | $\tau_{\max} = 7$ | $\tau_{\max} = 8$ | $\tau_{\max} = 9$ | $\tau_{\max} = 10$ |
|---|---|---|---|---|---|---|
| $F_1$-score$_{\text{all}}$ | $0.917 \pm 0.037$ | $0.910 \pm 0.038$ | $0.905 \pm 0.040$ | $0.902 \pm 0.042$ | $0.900 \pm 0.041$ | $0.897 \pm 0.042$ |
| SHD | $8.903 \pm 3.169$ | $9.280 \pm 3.289$ | $9.657 \pm 3.451$ | $9.953 \pm 3.492$ | $10.173 \pm 3.600$ | $10.423 \pm 3.568$ |

## 5 Conclusion

This paper addresses key challenges in causal discovery for autocorrelated time series by introducing the aMCI method and the ECD-aMCI algorithm. The proposed algorithm dynamically adapts conditioning sets to mitigate masking effects of strong autocorrelation while maintaining control over false discovery rates. Theoretical analysis establishes consistency, order independence, and enhanced accuracy properties. Extensive evaluations on both simulated and benchmark datasets demonstrate significant improvements in detection power for causal links, especially lagged links. The algorithm exhibits robust performance across varying autocorrelation strengths, sample sizes, and dimensionalities without requiring careful hyperparameter tuning.

REPRODUCIBILITY STATEMENT

The code to faithfully reproduce the results is provided in the Supplementary Material, and the model parameters and algorithm hyperparameters required for reproducing the results are included in the Appendix A.5.

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

## A  APPENDIX

### A.1  NOTATIONS

Table 3 summarizes the key mathematical notations used throughout the Appendix.

### A.2  BASELINE SELECTION CRITERIA

Causal discovery algorithms are typically categorized into four categories: constraint-based (CB), score-based (SB), structural causal model (SCM)-based, and Granger causality (GC)-based algorithms. The goal of the proposed algorithm is to learn causal graphs from observational time series that include both lagged (Window) and contemporaneous links. As shown in Table 4, algorithms that share this

Table 3: Summary of mathematical notations

| Notation | Description |
|---|---|
| $\mathcal{P}(X_{j,t})$ | True parent set of $X_{j,t}$ in $\mathcal{G}$ |
| $\hat{\mathcal{B}}^{-}(X_{j,t})$ | Estimated lagged parent set of $X_{j,t}$ |
| $\mathcal{B}^{-}(X_{j,t})$ | True lagged parent set of $X_{j,t}$ |
| $\hat{\mathcal{A}}(X_{j,t})$ | Estimated contemporaneous adjacency set of $X_{j,t}$ |
| $\mathcal{A}(X_{j,t})$ | True contemporaneous adjacency set of $X_{j,t}$ |
| $aMCI(\cdot)$ | Adaptive Momentary Conditional Independence method |
| $\perp\!\!\!\perp$ | Conditional independence relation |
| $\star-\star$ | Generic link (directed $\rightarrow$ or undirected $\circ-\circ$) |
| $\times-\times$ | Conflicting link orientation |
| $d$-separation | Graph-theoretic conditional independence criterion |

objective include PCMCI+ (Runge, 2020), PCMCI+Bagged (Debeire et al., 2024), DYNOTEARS Pamfil et al. (2020), NTS-NOTEARS (Sun et al., 2023), and VAR-LiNGAM (Hyvärinen et al., 2010). Among these, PCMCI+, PCMCI+Bagged, and NTS-NOTEARS are selected as baselines. DYNOTEARS and VAR-LiNGAM were not included because PCMCI+ has been demonstrated to outperform VAR-LiNGAM in its original paper, and NTS-NOTEARS has been shown to outperform DYNOTEARS in comparative studies.

Table 4: Summary of causal discovery algorithms for time series

| Category | Method | Window | Contemporaneous |
|---|---|---|---|
| CB | **PCMCI+** | Y | Y |
| CB | **Bagged-PCMCI+** | Y | Y |
| SB | **NTS-NOTEARS** | Y | Y |
| SB | DYNOTEARS | Y | Y |
| SCM | VAR-LiNGAM | Y | Y |
| SCM | TiMINo (Peters et al., 2013) | N | Y |
| SCM | NBCB (Assaad et al., 2021) | N | Y |
| SCM | NCDH (Wu et al., 2022) | N | N |
| GC | ACD (Löwe et al., 2022) | N | N |
| GC | CR-VAE (Li et al., 2023) | N | Y |

### A.3 PSEUDOCODES OF THE ECD-AMCI ALGORITHM

Section A.3 provides the complete set of pseudocodes for the aMCI method and the ECD-aMCI algorithm. Algorithm 1 presents the detailed procedures of the aMCI method. Algorithm 2 describes the PC$_1$ algorithm (**Phase 1**) in detail. Algorithm 3 describes the procedures of refining lagged parent sets via aMCI method (**Phase 2**). Algorithm 4 introduces the skeleton discovery procedure for both lagged and contemporaneous links (**Phase 3**). Algorithms 5-6 are the pseudo-codes for the collider phase and the orientation phase.

---

**Algorithm 1** The aMCI method

---

**Require:** Data $\boldsymbol{X}$ which is $d$-dimensional time series of length $T$, condition variable set $\mathcal{S}$, conditional independence test $CI(X_{i,t-\tau}, X_{j,t}, \mathcal{S})$ which returns $p$-value and test statistic value $I$, estimated lagged parent sets $\hat{\mathcal{B}}^-(X_{i,t-\tau})$ and $\hat{\mathcal{B}}^-(X_{j,t})$, confidence level $\alpha$

1: **if** $\tau > 0$ **then**
2:     **if** $X_{i,t-\tau-1} \in \hat{\mathcal{B}}^-(X_{i,t-\tau}) \cap \hat{\mathcal{B}}^-(X_{j,t})$ **then**
3:         $(p\text{-value}, I) \leftarrow CI(X_{i,t-\tau}, X_{j,t}, \mathcal{S} \cup \hat{\mathcal{B}}^-(X_{i,t-\tau}) \cup \hat{\mathcal{B}}^-(X_{j,t}))$
4:         $\mathcal{S}_{ad} \leftarrow \mathcal{S} \cup \hat{\mathcal{B}}^-(X_{i,t-\tau}) \cup \hat{\mathcal{B}}^-(X_{j,t})$
5:     **else**
6:         $\hat{\mathcal{B}}_{ad}^-(X_{i,t-\tau}) \leftarrow$ Substitute $X_{i,t-\tau-1}$ in $\hat{\mathcal{B}}^-(X_{i,t-\tau})$ with $X_{i,t-\tau-2}$
7:         $(p\text{-value}, I) \leftarrow CI(X_{i,t-\tau}, X_{j,t}, \mathcal{S} \cup \hat{\mathcal{B}}_{ad}^-(X_{i,t-\tau}) \cup \hat{\mathcal{B}}^-(X_{j,t}))$
8:         $\mathcal{S}_{ad} \leftarrow \mathcal{S} \cup \hat{\mathcal{B}}_{ad}^-(X_{i,t-\tau}) \cup \hat{\mathcal{B}}^-(X_{j,t})$
9:     **end if**
10: **else if** $\tau = 0$ **then**
11:     **if** both $X_{i,t-1}$ and $X_{j,t-1}$ are confounders of $(X_{i,t}, X_{j,t})$ **then**
12:         $(p\text{-value}, I) \leftarrow CI(X_{i,t}, X_{j,t}, \mathcal{S} \cup \hat{\mathcal{B}}^-(X_{i,t}) \cup \hat{\mathcal{B}}^-(X_{j,t}))$
13:         $\mathcal{S}_{ad} \leftarrow \mathcal{S} \cup \hat{\mathcal{B}}^-(X_{i,t}) \cup \hat{\mathcal{B}}^-(X_{j,t})$
14:     **else if** $X_{i,t-1}$ or $X_{j,t-1}$ is confounder of $(X_{i,t}, X_{j,t})$ **then**
15:         **if** $X_{i,t-1}$ is confounder of $(X_{i,t}, X_{j,t})$ **then**
16:             $\hat{\mathcal{B}}_{ad}^-(X_{j,t}) \leftarrow$ Substitute $X_{j,t-1}$ in $\hat{\mathcal{B}}^-(X_{j,t})$ with $X_{j,t-2}$
17:             $(p\text{-value}, I) \leftarrow CI(X_{i,t}, X_{j,t}, \mathcal{S} \cup \hat{\mathcal{B}}^-(X_{i,t}) \cup \hat{\mathcal{B}}_{ad}^-(X_{j,t}))$
18:             $\mathcal{S}_{ad} \leftarrow \mathcal{S} \cup \hat{\mathcal{B}}^-(X_{i,t}) \cup \hat{\mathcal{B}}_{ad}^-(X_{j,t})$
19:         **else if** $X_{j,t-1}$ is confounder of $(X_{i,t}, X_{j,t})$ **then**
20:             $\hat{\mathcal{B}}_{ad}^-(X_{i,t}) \leftarrow$ Substitute $X_{i,t-1}$ in $\hat{\mathcal{B}}^-(X_{i,t})$ with $X_{i,t-2}$
21:             $(p\text{-value}, I) \leftarrow CI(X_{i,t}, X_{j,t}, \mathcal{S} \cup \hat{\mathcal{B}}_{ad}^-(X_{i,t}) \cup \hat{\mathcal{B}}^-(X_{j,t}))$
22:             $\mathcal{S}_{ad} \leftarrow \mathcal{S} \cup \hat{\mathcal{B}}_{ad}^-(X_{i,t}) \cup \hat{\mathcal{B}}^-(X_{j,t})$
23:         **end if**
24:     **else if** Neither of $X_{i,t-1}$ and $X_{j,t-1}$ is confounder of $(X_{i,t}, X_{j,t})$ **then**
25:         $\hat{\mathcal{B}}_{ad}^-(X_{i,t}) \leftarrow$ Substitute $X_{i,t-1}$ in $\hat{\mathcal{B}}^-(X_{i,t})$ with $X_{i,t-2}$
26:         $(p\text{-value}_1, I_1) \leftarrow CI(X_{i,t}, X_{j,t}, \mathcal{S} \cup \hat{\mathcal{B}}_{ad}^-(X_{i,t}) \cup \hat{\mathcal{B}}^-(X_{j,t}))$
27:         $\mathcal{S}_{ad,1} \leftarrow \mathcal{S} \cup \hat{\mathcal{B}}_{ad}^-(X_{i,t}) \cup \hat{\mathcal{B}}^-(X_{j,t})$
28:         $\hat{\mathcal{B}}_{ad}^-(X_{j,t}) \leftarrow$ Substitute $X_{j,t-1}$ in $\hat{\mathcal{B}}^-(X_{j,t})$ with $X_{j,t-2}$
29:         $(p\text{-value}_2, I_2) \leftarrow CI(X_{i,t}, X_{j,t}, \mathcal{S} \cup \hat{\mathcal{B}}^-(X_{i,t}) \cup \hat{\mathcal{B}}_{ad}^-(X_{j,t}))$
30:         $\mathcal{S}_{ad,2} \leftarrow \mathcal{S} \cup \hat{\mathcal{B}}^-(X_{i,t}) \cup \hat{\mathcal{B}}_{ad}^-(X_{j,t})$
31:         $(p\text{-value}_3, I_3) \leftarrow CI(X_{i,t}, X_{j,t}, \mathcal{S} \cup \hat{\mathcal{B}}^-(X_{i,t}) \cup \hat{\mathcal{B}}^-(X_{j,t}))$
32:         $\mathcal{S}_{ad,3} \leftarrow \mathcal{S} \cup \hat{\mathcal{B}}^-(X_{i,t}) \cup \hat{\mathcal{B}}^-(X_{j,t})$
33:         $(p\text{-value}, I, \mathcal{S}_{ad}) \leftarrow (p\text{-value}_{k^*}, I_{k^*}, \mathcal{S}_{ad,k^*})$     where    $k^* = \underset{k \in \{1,2,3\}}{\arg\min}\, p\text{-value}_k$
34:     **end if**
35: **end if**
36: **return** $(p\text{-value}, I, \mathcal{S}_{ad})$ for arbitrary $(X_{i,t-\tau}, X_{j,t}, \mathcal{S})$

---

---

**Algorithm 2** Phase 1: $PC_1$-Based Initial Estimation of Lagged Parent Sets

---

**Require:** Time series dataset $\boldsymbol{X} := \{\boldsymbol{X_t} \mid t \in \{1, \ldots, T\}\}$, maximum lag $\tau_{\max}$, confidence level $\alpha$, conditional independence test $CI(X_{i,t-\tau}, X_{j,t}, \mathcal{S})$ which returns $p$-value and test statistic value $I$.

1: **for all** $j$ in $\{1, \ldots, d\}$ **do**
2:     Initialize lagged parent set $\hat{\mathcal{B}}^-(X_{j,t}) \leftarrow (X_{t-1}^j, \ldots, X_{t-\tau_{\max}}^j)$ and min test statistic values $I_{min}(X_{i,t-\tau}, X_{j,t}) \leftarrow \infty$ for all $X_{i,t-\tau} \in \hat{\mathcal{B}}^-(X_{j,t})$.
3:     Let $p \leftarrow 0$.
4:     **while** $p \leq 1$ **and** any $X_{i,t-\tau} \in \hat{\mathcal{B}}^-(X_{j,t})$ satisfies $|\hat{\mathcal{B}}^-(X_{j,t}) \setminus \{X_{i,t-\tau}\}| \geq p$ **do**
5:         **for all** $X_{i,t-\tau}$ in $\hat{\mathcal{B}}^-(X_{j,t})$ satisfying $|\hat{\mathcal{B}}^-(X_{j,t}) \setminus \{X_{i,t-\tau}\}| \geq p$ **do**
6:             $\mathcal{S} =$ first $p$ variables in $\hat{\mathcal{B}}^-(X_{j,t}) \setminus \{X_{i,t-\tau}\}$.          ▷ Select conditioning set of size $p$
7:             $(p\text{-value}, I) \leftarrow CI(X_{i,t-\tau}, X_{j,t}, \mathcal{S})$
8:             $I_{min}(X_{i,t-\tau}, X_{j,t}) = \min(I, I_{min}(X_{i,t-\tau}, X_{j,t}))$
9:             **if** $p$-value $> \alpha$ **then**
10:                Mark $X_{i,t-\tau}$ for removal from $\hat{\mathcal{B}}^-(X_{j,t})$.
11:            **end if**
12:        **end for**
13:        Remove non-significant entries from $\hat{\mathcal{B}}^-(X_{j,t})$ and sort remaining entries in $\hat{\mathcal{B}}^-(X_{j,t})$ by $I_{min}(X_{i,t-\tau}, X_{j,t})$ from largest to smallest.
14:        Let $p \leftarrow p + 1$.
15:     **end while**
16: **end for**
17: **return** $\hat{\mathcal{B}}^-(X_{j,t})$ for all $j \in \{1, \ldots, d\}$          ▷ The estimated lagged parent sets

---

**Algorithm 3** Phase 2: Refined Lagged Parent Skeleton via aMCI

---

**Require:** Time series dataset $\boldsymbol{X}$, maximum lag $\tau_{\max}$, confidence level $\alpha$, aMCI criterion $aMCI(X_{i,t-\tau}, X_{j,t}, \mathcal{S}, \hat{\mathcal{B}}^-(X_{i,t-\tau}), \hat{\mathcal{B}}^-(X_{j,t}))$ which returns $p$-value, test statistic value $I$ and adaptive conditioning set $\mathcal{S}_{ad}$, estimated lagged parent sets $\hat{\mathcal{B}}^-(X_{j,t})$ for all $j$ in $\{1, \ldots, d\}$, confidence level $\alpha$

1: **Initialize** graph $\hat{\mathcal{G}}$ with fully connected lagged links (up to $\tau_{\max}$) and contemporaneous links
2: **Initialize** contemporaneous adjacency sets $\hat{\mathcal{A}}(X_{j,t}) \leftarrow \{X_{k,t} \mid k \in \{1, \ldots, d\}, k \neq j\}$ for all $j \in \{1, \ldots, d\}$.
3: $p \leftarrow 0$
4: **while** $p \leq 1$ **and** any adjacent pair $(X_{i,t-\tau}, X_{j,t})$ for $0 \leq \tau \leq \tau_{\max}$ in $\hat{\mathcal{G}}$ satisfies $|\hat{\mathcal{A}}(X_{i,t-\tau}) \setminus \{X_{j,t}\}| \geq p$ or $|\hat{\mathcal{A}}(X_{j,t}) \setminus \{X_{i,t-\tau}\}| \geq p$ **do**
5:     **for all** adjacent pairs $(X_{i,t-\tau}, X_{j,t})$ for $0 < \tau \leq \tau_{\max}$ satisfying the condition in Line 5 **do**
6:         **for all** possible subsets $\mathcal{S} \subseteq \hat{\mathcal{A}}(X_{j,t})$ with $|\mathcal{S}| = p$ **do**
7:             $(p\text{-value}, I, \mathcal{S}_{ad}) \leftarrow aMCI(X_{i,t-\tau}, X_{j,t}, \mathcal{S}, \hat{\mathcal{B}}^-(X_{i,t-\tau}), \hat{\mathcal{B}}^-(X_{j,t}))$
8:             **if** $p$-value $> \alpha$ **then**
9:                 Delete link $X_{i,t-\tau} \rightarrow X_{j,t}$ in $\hat{\mathcal{G}}$ for $\tau > 0$ from $\hat{\mathcal{G}}$
10:                Store $\mathcal{S}_{ad}$ as seperating set of $(X_{i,t-\tau}, X_{j,t})$
11:            **end if**
12:        **end for**
13:    **end for**
14:    Update $\hat{\mathcal{B}}^-(X_{j,t})$ for $j$ in $\{1, \ldots, d\}$ based on estimated $\hat{\mathcal{G}}$
15:    $p \leftarrow p + 1$
16: **end while**
17: **return** $\hat{\mathcal{B}}^-(X_{j,t})$ for all $j \in \{1, \ldots, d\}$

---

---

**Algorithm 4** Phase 3: Complete Skeleton Discovery

---

**Require:** Time series dataset $\boldsymbol{X}$, maximum lag $\tau_{\max}$, confidence level $\alpha$, aMCI method $aMCI(X_{i,t-\tau}, X_{j,t}, \mathcal{S}, \hat{\mathcal{B}}^-(X_{i,t-\tau}), \hat{\mathcal{B}}^-(X_{j,t}))$ which returns $p$-value, test statistic value $I$ and adaptive conditioning set $\mathcal{S}_{ad}$, estimated lagged parent sets $\hat{\mathcal{B}}^-(X_{j,t})$ for all $j$ in $\{1, \ldots, d\}$, confidence level $\alpha$

1: **Initialize** graph $\hat{\mathcal{G}}$ with fully connected lagged links (up to $\tau_{\max}$) and contemporaneous links
2: **Initialize** contemporaneous adjacency sets $\hat{\mathcal{A}}(X_{j,t}) \leftarrow \{X_{k,t} \mid k \in \{1, \ldots, d\}, k \neq j\}$ for all $j \in \{1, \ldots, d\}$.
3: **Initialize** $I_{min}(X_{i,t-\tau}, X_{j,t}) \leftarrow \infty$ for all links in $\hat{\mathcal{G}}$
4: $p \leftarrow 0$
5: **while** any adjacent pair $(X_{i,t-\tau}, X_{j,t})$ for $0 \leq \tau \leq \tau_{\max}$ in $\hat{\mathcal{G}}$ satisfies $|\hat{\mathcal{A}}(X_{i,t-\tau}) \setminus \{X_{j,t}\}| \geq p$ or $|\hat{\mathcal{A}}(X_{j,t}) \setminus \{X_{i,t-\tau}\}| \geq p$ **do**
6:    **for all** adjacent pairs $(X_{i,t-\tau}, X_{j,t})$ for $0 < \tau \leq \tau_{\max}$ satisfying the condition in Line 5 **do**
7:       **for all** possible subsets $\mathcal{S} \subseteq \hat{\mathcal{A}}(X_{j,t})$ with $|\mathcal{S}| = p$ **do**
8:          $(p\text{-value}, I, \mathcal{S}_{ad}) \leftarrow aMCI(X_{i,t-\tau}, X_{j,t}, \mathcal{S}, \hat{\mathcal{B}}^-(X_{i,t-\tau}), \hat{\mathcal{B}}^-(X_{j,t}))$
9:          **if** $p$-value $> \alpha$ **then**
10:             Delete link $X_{i,t-\tau} \rightarrow X_{j,t}$ in $\hat{\mathcal{G}}$ for $\tau > 0$ from $\hat{\mathcal{G}}$
11:             Store $\mathcal{S}_{ad}$ as seperating set of $(X_{i,t-\tau}, X_{j,t})$
12:          **end if**
13:       **end for**
14:    **end for**
15:    Update $\hat{\mathcal{B}}(X_{j,t})$ for $j$ in $\{1, \ldots, d\}$ based on estimated $\hat{\mathcal{G}}$
16:    **for all** adjacent pairs $(X_{i,t}, X_{j,t})$ satisfying the condition in Line 5 **do**
17:       **for all** possible subsets $\mathcal{S} \subseteq \hat{\mathcal{A}}(X_{j,t}) \setminus \{X_{i,t}\}$ with $|\mathcal{S}| = p$ **do**
18:          $(p\text{-value}, I, \mathcal{S}_{ad}) \leftarrow aMCI(X_{i,t}, X_{j,t}, \mathcal{S}, \hat{\mathcal{B}}^-(X_{i,t}), \hat{\mathcal{B}}^-(X_{j,t}))$
19:          **if** $p$-value $> \alpha$ **then**
20:             Delete link $X_{i,t} \leftrightarrow X_{j,t}$ from $\hat{\mathcal{G}}$
21:             Store $\mathcal{S}_{ad}$ as seperating set of $(X_{i,t}, X_{j,t})$
22:          **end if**
23:       **end for**
24:    **end for**
25:    Update $\hat{\mathcal{A}}(X_{j,t})$ for $j$ in $\{1, \ldots, d\}$ based on estimated $\hat{\mathcal{G}}$
26:    $p \leftarrow p + 1$
27: **end while**
28: **return** Graph $\hat{\mathcal{G}}$, Separating sets of all nonadjacent pairs.

---

**Algorithm 5** Detailed collider phase with conservative rules

---

**Require:** $\hat{\mathcal{G}}$ and separating sets from Algorithm 4, time series dataset $\boldsymbol{X}$, confidence level $\alpha$, $CI(X, Y, Z)$, $\hat{\mathcal{B}}^-(X_{j,t})$ for all $j$ in $\{1, \ldots, d\}$.

1: **for** all unshielded triples $X_{i,t-\tau} \rightarrow X_{k,t} \circ\!\!-\!\!\circ X_{j,t}$ ($\tau > 0$) or $X_{i,t} \circ\!\!-\!\!\circ X_{k,t} \circ\!\!-\!\!\circ X_{j,t}$ ($\tau = 0$) in $\hat{\mathcal{G}}$ where $(X_{i,t-\tau}, X_{j,t})$ are not adjacent **do**
2:    Define contemporaneous adjacencies $\hat{\mathcal{A}}(X_{j,t}) \leftarrow \{X_{i,t} \neq X_{j,t} \in \boldsymbol{X_t} : X_{i,t} \circ\!\!-\!\!\circ X_{j,t}$ in $\hat{\mathcal{G}}\}$
3:    **for all** $\mathcal{S} \subseteq \hat{\mathcal{A}}(X_{j,t}) \setminus \{X_{i,t-\tau}\}$ and for all $\mathcal{S} \subseteq \hat{\mathcal{A}}(X_{i,t}) \setminus \{X_{j,t}\}$ (if $\tau = 0$) **do**
4:       $(p\text{-value}, I, \mathcal{S}) \leftarrow CI(X_{i,t-\tau}, X_{j,t}, S \cup \hat{\mathcal{B}}^-(X_{j,t}) \cup \hat{\mathcal{B}}^-(X_{i,t-\tau}) \setminus \{X_{i,t-\tau}\})$
5:       Store subset $\mathcal{S}$ with $p$-value $> \alpha$ as separating subset
6:    **end for**
7:    **if** no separating subsets are found **then**
8:       Mark triple as ambiguous
9:    **else**
10:       Compute fraction $n_k$ of separating subsets that contain $X_{k,t}$, orient triple as collider if $n_k = 0$, leave unoriented if $n_k = 1$, and mark as ambiguous if $0 < n_k < 1$
11:    **end if**
12:    Mark links in $\hat{\mathcal{G}}$ with conflicting orientations as $\times\!-\!\times$
13: **end for**
14: **return** $\hat{\mathcal{G}}$, seperating set, ambiguous triples, conflicting links

---

---

**Algorithm 6** Detailed rule orientation phase

---

**Require:** $\hat{\mathcal{G}}$, ambiguous triples, conflicting links
1: **while** any unambiguous triples suitable for rules R1-R3 are remaining **do**
2:      Apply rule R1 (orient unshielded triples that are not colliders):
3:      **for** all unambiguous triples $X_{i,t-\tau} \rightarrow X_{k,t} \circ-\circ X_{j,t}$ where $(X_{i,t-\tau}, X_{j,t})$ are not adjacent **do**
4:          Orient as $X_{i,t-\tau} \rightarrow X_{k,t} \rightarrow X_{j,t}$
5:      **end for**
6:      Mark links with conflicting orientations as $\times-\times$
7:      Apply rule R2 (avoid cycles):
8:      **for** all unambiguous triples $X_{i,t} \rightarrow X_{k,t} \rightarrow X_{j,t}$ with $X_{i,t} \circ-\circ X_{j,t}$ **do**
9:          Orient as $X_{i,t} \rightarrow X_{j,t}$
10:     **end for**
11:     Mark links with conflicting orientations as $\times-\times$
12:     Apply rule R3 (orient unshielded triples that are not colliders and avoid cycles):
13:     **for** all pairs of unambiguous triples $X_{i,t} \circ-\circ X_{k,t} \rightarrow X_{j,t}$ and $X_{i,t} \circ-\circ X_{l,t} \rightarrow X_{j,t}$ where $(X_{k,t}, X_{l,t})$ are not adjacent and $X_{i,t} \circ-\circ X_{j,t}$ **do**
14:         Orient as $X_{i,t} \rightarrow X_{j,t}$
15:     **end for**
16:     Mark links with conflicting orientations as $\times-\times$
17: **end while**
18: **return** $\hat{\mathcal{G}}$, conflicting links

---

## A.4 THEORETICAL ANALYSIS

### A.4.1 DEFINITION OF ASSUMPTIONS

The theoretical analysis relies on several standard assumptions in causal discovery: (A1) Causal Sufficiency, (A2) Causal Markov Condition, (A3) Faithfulness, (A4) Temporal Priority, and (A5) Stationarity. These assumptions are commonly adopted in the causal discovery literature (Spirtes et al., 2001; Runge et al., 2019; Runge, 2020; Biswas & Shlizerman, 2022).

**Assumption A1 (Causal Sufficiency)** All common causes of any pair of observed variables in the system are also observed.

**Assumption A2 (Causal Markov Condition)** Each variable in the causal graph is conditionally independent of its non-descendants given its direct parents. Formally, if $\mathcal{G}$ is a directed acyclic graph (DAG) representing the causal structure, and $P$ is the joint probability distribution of the variables, then $P$ factorizes according to $\mathcal{G}$ as:

$$P(X_1, X_2, \ldots, X_d) = \prod_{i=1}^{d} P(X_i | \mathcal{P}(X_i)),$$

where $\mathcal{P}(X_i)$ denotes the parent set of $X_i$ in $\mathcal{G}$.

**Assumption A3 (Faithfulness)** All conditional independence relationships in the probability distribution are entailed by the causal graph structure via the d-separation criterion. That is, $X \perp\!\!\!\perp Y \mid \mathcal{Z}$ if and only if $X$ and $Y$ are d-separated by $\mathcal{Z}$ in the causal graph. This assumption rules out precise parameter cancellations that could create conditional independencies not implied by the causal structure.

**Assumption A4 (Temporal Priority)** A cause precedes its effect in time, or at the very least, occurs simultaneously. Furthermore, as the time lag increases, the autoregressive influence of a process on its own future values progressively diminishes.

**Assumption A5 (Stationarity)** The causal structure among variables and the population of time series do not change over time. This assumption allows us to learn a window causal graph from time series, where edges represent consistent causal links across all time points.

### A.4.2 PROOFS OF THEORETICAL PROPERTIES

**Theorem 4 (Consistency)** *Under the Assumptions (A1)-(A5), if the conditional independence tests are oracle, ECD-aMCI returns the correct CPDAG, i.e., $\hat{\mathcal{G}} = \mathcal{G}_{CPDAG}$, where $\mathcal{G}_{CPDAG}$ denotes the CPDAG of the time series graph $\mathcal{G}$.*

**Proof** The proof of consistency comprises three primary steps: initially demonstrating that Algorithms 2-3 return a superset of the lagged parent set $\hat{\mathcal{B}}^-(X_{j,t})$, then establishing that Algorithm 4 accurately recovers the skeleton of the causal graph, and finally establishing that all unshielded triples that are colliders are correctly identified.

Step 1: Superset Property of Estimated Lagged Parents. Since Algorithm 2 is adopted from Algorithm 1 in Runge (2020), the estimated lagged parent set $\hat{\mathcal{B}}^-(X_{j,t})$ satisfies the condition $\mathcal{B}^-(X_{j,t}) \subseteq \hat{\mathcal{B}}^-(X_{j,t})$ based on Lemma S1 in Runge (2020). Given oracle conditional independence tests, we assert that the outcome of the aMCI method is equivalent to that of MCI. Without loss of generality, we only need to prove that if $X_{i,t-\tau-1}$ is not a confounder, then replacing $X_{i,t-\tau-1}$ with $X_{i,t-\tau-2}$ does not affect the conditional independence conclusion—that is, testing $X_{i,t-\tau} \perp\!\!\!\perp X_{j,t} \mid X_{i,t-\tau-2}$ instead of $X_{i,t-\tau} \perp\!\!\!\perp X_{j,t} \mid X_{i,t-\tau-1}$ still yields the same result. Under the Causal Sufficiency and Faithfulness assumptions, $X \perp\!\!\!\perp Y \mid \mathbf{Z}$ if and only if $X$ is d-separated from $Y$ given $\mathbf{Z}$. Under the Temporal Priority Assumption, $X_{i,t-\tau-1}$ cannot be a collider. Therefore, if $X_{i,t-\tau-1}$ is also not a confounder, then according to the definition of d-separation in Section 2, replacing $X_{i,t-\tau-1}$ with $X_{i,t-\tau-2}$ does not affect the d-separation between $X_{i,t-\tau}$ and $X_{j,t}$. That is, $X_{i,t-\tau} \perp\!\!\!\perp X_{j,t} \mid X_{i,t-\tau-1}$ if and only if $X_{i,t-\tau} \perp\!\!\!\perp X_{j,t} \mid X_{i,t-\tau-2}$. Since the two conditional independence statements are equivalent and the tests are assumed to be oracle, the outcome of the aMCI method is equivalent to that of MCI. Consequently, Algorithm 3 does not update the originally estimated lagged parents at $p = 0$. In the $p = 1$ phase, Algorithm 3 will remove parents of one contemporaneous parent of $X_{j,t}$ that are not parents of $X_{j,t}$ directly. Therefore, the refined estimated lagged parent set remains a superset of the true lagged parent set $\mathcal{B}^-(X_{j,t})$.

Step 2: Correctness of the Skeleton Discovery. In this step, $\hat{\mathcal{G}}^* = \mathcal{G}^*$ is proved under Assumptions (A1)-(A5), where $\mathcal{G}^*$ represents the skeleton of the causal graph. To establish this equality, it is sufficient to demonstrate that for any arbitrary $X_{i,t-\tau}, X_{j,t}$, the following statements hold: (1) $X_{i,t-\tau} \star\!\!-\!\!\star X_{j,t} \notin \hat{\mathcal{G}}^* \Rightarrow X_{i,t-\tau} \star\!\!-\!\!\star X_{j,t} \notin \mathcal{G}^*$ and (2) $X_{i,t-\tau} \star\!\!-\!\!\star X_{j,t} \notin \mathcal{G}^* \Rightarrow X_{i,t-\tau} \star\!\!-\!\!\star X_{j,t} \notin \hat{\mathcal{G}}^*$, where $\star\!\!-\!\!\star$ denotes both types of skeleton links (directed links $\to$ and undirected links $\circ\!\!-\!\!\circ$) for simplicity.

(1) Algorithm 2 eliminates a link $X_{i,t-\tau} \star\!\!-\!\!\star X_{j,t}$ from $\hat{\mathcal{G}}^*$ if and only if $X_{i,t-\tau} \perp\!\!\!\perp X_{j,t} \mid \mathcal{S}_{ad}$ for some subset $\mathcal{S} \subseteq \hat{\mathcal{A}}(X_{j,t})$ during the iterative conditional independence tests, where $\mathcal{S}_{ad}$ is the third outcome of the aMCI. Here, $\hat{\mathcal{A}}(X_{j,t})$ denotes the contemporaneous adjacencies. By the principle of Faithfulness, this conditional independence directly implies that $X_{i,t-\tau}$ and $X_{j,t}$ are d-seperated by $\mathcal{S}_{ad}$ in the true causal graph, thus $X_{i,t-\tau} \star\!\!-\!\!\star X_{j,t} \notin \mathcal{G}^*$.

(2) According to the conclusion of Step 1, $\hat{\mathcal{B}}^-(X_{i,t-\tau}) \cup \hat{\mathcal{B}}^-(X_{j,t}) \setminus X_{i,t-\tau}$ does not contain descendants. Thus, the Causal Markov Condition yields $(X_{i,t-\tau}, \hat{\mathcal{B}}^-(X_{i,t-\tau}) \cup \hat{\mathcal{B}}^-(X_{j,t}) \setminus X_{i,t-\tau}) \perp\!\!\!\perp X_{j,t} \mid \mathcal{P}(X_{j,t})$. Applying the weak union property of conditional independence, we derive $X_{i,t-\tau} \perp\!\!\!\perp X_{j,t} \mid \mathcal{P}(X_{j,t}) \cup (\hat{\mathcal{B}}^-(X_{i,t-\tau}) \cup \hat{\mathcal{B}}^-(X_{j,t}) \setminus X_{i,t-\tau})$. Note that $X_{i,t-\tau} \notin \mathcal{P}(X_{j,t})$ since $X_{i,t-\tau} \star\!\!-\!\!\star X_{j,t} \notin \mathcal{G}^*$, for the case where $\tau = 0$, we assume $X_{i,t}$ is not a descendant of $X_{j,t}$ (as the alternate case would be covered by exchanging $X_{i,t}$ and $X_{j,t}$).

Now it suffices to prove that the conditioning set $\mathcal{P}(X_{j,t}) \cup (\hat{\mathcal{B}}^-(X_{i,t-\tau}) \cup \hat{\mathcal{B}}^-(X_{j,t}) \setminus X_{i,t-\tau})$ must be tested in Algorithm 4. Algorithm 4 systematically tests $X_{i,t-\tau} \perp\!\!\!\perp X_{j,t} \mid \mathcal{S} \cup \hat{\mathcal{B}}^-(X_{i,t-\tau}) \cup \hat{\mathcal{B}}^-(X_{j,t}) \setminus X_{i,t-\tau}$ across all subsets $\mathcal{S} \subseteq \hat{\mathcal{A}}(X_{j,t})$. The contrapositive of Step 2 (1) confirms that the estimated contemporaneous adjacencies consistently include the true contemporaneous adjacencies as a subset, i.e., $\mathcal{A}(X_{j,t}) \subseteq \hat{\mathcal{A}}(X_{j,t})$. Furthermore, Step 1 also confirms that $\hat{\mathcal{B}}^-(X_{j,t})$ encompasses all lagged parents of $X_{j,t}$, i.e., $\mathcal{B}^-(X_{j,t}) \subseteq \hat{\mathcal{B}}^-(X_{j,t})$. Consequently, during the iteration process, there exists a subset $\mathcal{S}$ such that $\mathcal{S} \cup \hat{\mathcal{B}}^-(X_{i,t-\tau}) \cup \hat{\mathcal{B}}^-(X_{j,t}) \setminus X_{i,t-\tau} = \mathcal{P}(X_{j,t}) \cup (\hat{\mathcal{B}}^-(X_{i,t-\tau}) \cup \hat{\mathcal{B}}^-(X_{j,t}) \setminus X_{i,t-\tau})$.

Algorithm 4 will detect $X_{i,t-\tau} \perp\!\!\!\perp X_{j,t} \mid \mathcal{P}(X_{j,t}) \cup (\hat{\mathcal{B}}^-(X_{i,t-\tau}) \cup \hat{\mathcal{B}}^-(X_{j,t}) \setminus X_{i,t-\tau})$ and subsequently remove $X_{i,t-\tau} \star\!-\!\star X_{j,t}$ from $\hat{\mathcal{G}}^*$.

Step 3: The proof that all unshielded triples that are colliders are correctly identified involves two aspects: (1) If unshielded triples are oriented as colliders in Algorithm 5, then these triples are truly colliders (establishing the correctness of the collider orientation phase), and (2) All unshielded triples that are colliders can be correctly identified. Based on the established correctness of the skeleton discovery in Step 2 and the reliability of oracle conditional independence tests, the triples that are oriented as colliders in Line 10 of Algorithm 5 are correct. Therefore, it remains to show that all unshielded triples that are colliders are correctly identified.

Considering a generic triple $X_{i,t_i} \star\!-\!\star X_{k,t_k} \star\!-\!\star X_{j,t_j}$, we can fix $t_j = t$ by stationarity. Time order constraints and stationarity properties allow us to reduce the analysis to two specific cases: $X_{i,t-\tau} \to X_{k,t} \circ\!-\!\circ X_{j,t}$ (for $\tau > 0$) or $X_{i,t} \circ\!-\!\circ X_{k,t} \circ\!-\!\circ X_{j,t}$ (for $\tau = 0$) in $\mathcal{G}$ where $X_{i,t-\tau}$ and $X_{j,t}$ are not adjacent. Since $X_{k,t}$ is contemporaneous with $X_{j,t}$, only contemporaneous components of separating sets are relevant for the collider orientation phase. Given the correctness of the skeleton discovery and the fact that $(X_{i,t-\tau}, X_{j,t})$ are not adjacent, there exists a subset $\mathcal{S}$ such that $X_{i,t-\tau} \perp\!\!\!\perp X_{j,t} \mid \mathcal{S} \cup \hat{\mathcal{B}}^-(X_{i,t-\tau}) \cup \hat{\mathcal{B}}^-(X_{j,t}) \setminus \{X_{i,t-\tau}\}$. Furthermore, by the Faithfulness assumption and the definition of d-separation, $X_{k,t}$ cannot belong to any set $\mathcal{S}$ for which $X_{i,t-\tau} \perp\!\!\!\perp X_{j,t} \mid \mathcal{S} \cup \hat{\mathcal{B}}^-(X_{i,t-\tau}) \cup \hat{\mathcal{B}}^-(X_{j,t}) \setminus \{X_{i,t-\tau}\}$. This implies that all unshielded triples that are colliders can be correctly identified according to the rules of the Algorithm 5,.

**Theorem 5 (Enhanced Accuracy)** *Under the Assumptions (A1)-(A5), applying the aMCI method yields a higher probability of learning the correct causal graph, i.e., $P(\hat{\mathcal{G}} = \mathcal{G}_{CPDAG}) \geq P(\tilde{\mathcal{G}} = \mathcal{G}_{CPDAG})$, where $\tilde{\mathcal{G}}$ represents the causal graph estimated without applying the aMCI method.*

**Proof** Let $\boldsymbol{A}$ be any alternative causal discovery algorithm not employing the aMCI method. $P(\hat{\mathcal{G}} = \mathcal{G}_{\text{CPDAG}}) \geq P(\tilde{\mathcal{G}} = \mathcal{G}_{\text{CPDAG}})$ is proved through two steps.

Step 1: When algorithm $\boldsymbol{A}$ correctly identifies the completed partially directed acyclic graph $\mathcal{G}_{\text{CPDAG}}$, the ECD-aMCI algorithm implements specific modifications to enhance causal discovery. For lagged links ($\tau > 0$), ECD-aMCI modifies the conditioning sets $\mathcal{S}$ specifically in cases where $X_{i,t-\tau-1} \notin \hat{\mathcal{B}}^-(X_{i,t-\tau}) \cap \hat{\mathcal{B}}^-(X_{j,t})$. For contemporaneous links ($\tau = 0$), the adaptive MCI (aMCI) component substitutes conditioning variables with more appropriate ones when the original variables are identified as non-confounders, based on the refined lagged parent set $\hat{\mathcal{B}}^-(X_{i,t})$ and $\hat{\mathcal{B}}^-(X_{j,t})$.

Due to the consistency properties of the ECD-aMCI algorithm, this approach preserves all necessary confounders in the conditioning set $\mathcal{S}$ while systematically removing immediate predecessors that are non-confounders. This selective conditioning strategy ensures that the estimated graph $\hat{\mathcal{G}}$ accurately converges to $\mathcal{G}_{\text{CPDAG}}$. Consequently, we can establish that $\{\tilde{\mathcal{G}} = \mathcal{G}_{\text{CPDAG}}\} \subseteq \{\hat{\mathcal{G}} = \mathcal{G}_{\text{CPDAG}}\}$.

Step 2: Considering the model (2) in paper with strong autocorrelation. For the causal link $X_{i,t-1} \to X_{j,t}$, standard algorithms $\boldsymbol{A}$ (e.g., PCMCI+) tests $X_{i,t-1} \perp\!\!\!\perp X_{j,t} \mid \{X_{i,t-2}, X_{j,t-1}\}$. As discussed in Section 3.1 in paper, this test almost relies on detecting the correlation between $\varepsilon_{i,t-1}$ and $\varepsilon_{j,t}$, which are independent. In contrast, ECD-aMCI algorithm substitutes $X_{i,t-2}$ with $X_{i,t-3}$ by applying the aMCI method, which transforms the test to one that relies on detecting the correlation between $\{\varepsilon_{i,t-1}, \varepsilon_{i,t-2}\}$ and $\{\varepsilon_{j,t}, \varepsilon_{i,t-1}, \varepsilon_{i,t-2}\}$, which exhibit dependency due to the shared terms. This strategic adaptation enables correct detection of the causal link $X_{i,t-1} \to X_{j,t}$ in scenarios where algorithm $\boldsymbol{A}$ fails. This demonstrates that there exist datasets for which $\hat{\mathcal{G}} = \mathcal{G}_{\text{CPDAG}}$ but $\tilde{\mathcal{G}} \neq \mathcal{G}_{\text{CPDAG}}$.

Combining these results, we have:

$$P(\hat{\mathcal{G}} = \mathcal{G}_{\text{CPDAG}}) = P(\hat{\mathcal{G}} = \mathcal{G}_{\text{CPDAG}}, \tilde{\mathcal{G}} = \mathcal{G}_{\text{CPDAG}}) + P(\hat{\mathcal{G}} = \mathcal{G}_{\text{CPDAG}}, \tilde{\mathcal{G}} \neq \mathcal{G}_{\text{CPDAG}})$$
$$= P(\tilde{\mathcal{G}} = \mathcal{G}_{\text{CPDAG}}) + P(\hat{\mathcal{G}} = \mathcal{G}_{\text{CPDAG}}, \tilde{\mathcal{G}} \neq \mathcal{G}_{\text{CPDAG}}),$$

where the second term is strictly positive based on the datasets mentioned in the last paragraph. This directly yields:

$$P(\hat{\mathcal{G}} = \mathcal{G}_{\text{CPDAG}}) \geq P(\tilde{\mathcal{G}} = \mathcal{G}_{\text{CPDAG}}),$$

completing the proof of Theorem 2.

**Theorem 6 (Order Independence)** *Under the Assumptions (A1)-(A5), the outcome of ECD-aMCI is independent of the order of the variables.*

**Proof** The proof of the order-independence consists of three main steps: first demonstrating that the aMCI method is order-independent, then proving that the estimation of refined lagged parent sets and complete skeleton is order-independent, and finally establishing the order-independence of the orientation process.

Step 1: Order-independence of the aMCI method. The aMCI method, by definition, operates as a mapping function that takes inputs $(X_{i,t-\tau}, X_{j,t}, \mathcal{S}, \hat{\mathcal{B}}^-(X_{i,t-\tau}), \hat{\mathcal{B}}^-(X_{j,t}))$ and produces outcomes determined solely by these inputs. Since the outcomes are completely determined by the input values rather than the order in which variables are processed, the application of the aMCI method preserves the order-independence property of any algorithm that incorporates it.

Step 2: Order-independence of parent set estimation and skeleton discovery. The processes described in Algorithms 2-4 maintain order-independence by adopting the idea of PC-stable. In Algorithms 2-4, edge removals are executed only after completing each iteration over conditioning sets of cardinality $p$. As established in Step 1, the integration of the aMCI method does not compromise this property, thereby ensuring that the estimation of refined lagged parent sets and the discovery of the complete skeleton remain order-independent.

Step 3: Order-independence of orientation phases. The orientation process maintains order-independence through careful management of potential conflicts. Following the methodology established in Runge (2020), the collider identification phase (Algorithm 5) and the rule-based orientation phase (Algorithm 6) preserve order-independence by implementing two key strategies: (1) ambiguity marking for triples with inconsistent separating sets, and (2) consistent marking of conflicting link orientations using the notation $\times - \times$.

### A.5 ADDITIONAL EVALUATION RESULTS AND HYPERPARAMETER SETTINGS

### A.5.1 DETAILED RESULTS AND HYPERPARAMETERS OF LINEAR SETTINGS (FIGURE 2)

First, the Tables 5-16 present the mean and standard error of all metrics from Figure 2 in the main paper. Then, all hyperparameters required for the experiments in Figure 2 are provided. The simulation experiments systematically evaluate performance across three parameters: autocorrelation strength ($a$), sample size ($T$), and dimension of the time series ($d$).

**Results:** According to the standard errors presented in Tables 5-16, the ECD-aMCI algorithm demonstrates the smallest standard errors for F1-score$_{lagged}$, F1-score$_{all}$ and SHD across most settings, indicating robust performance and significantly superior results compared to baseline algorithms.

Table 5: F1-score$_{lagged}$ Comparison ($a$ variation)

| Method | $a$ | | | | |
|---|---|---|---|---|---|
| | 0.9 | 0.8 | 0.7 | 0.6 | 0.5 |
| ECD-aMCI | **0.775 ± 0.106** | **0.848 ± 0.077** | **0.873 ± 0.072** | **0.884 ± 0.067** | **0.886 ± 0.064** |
| PCMCI+ | 0.519 ± 0.156 | 0.695 ± 0.118 | 0.807 ± 0.086 | 0.848 ± 0.073 | 0.867 ± 0.066 |
| Bagged-PCMCI+ | 0.456 ± 0.148 | 0.586 ± 0.121 | 0.672 ± 0.101 | 0.679 ± 0.098 | 0.671 ± 0.091 |
| NTS-NOTEARS | 0.240 ± 0.150 | 0.493 ± 0.194 | 0.708 ± 0.097 | 0.799 ± 0.074 | 0.837 ± 0.071 |

Table 6: F1-score$_{all}$ Comparison ($a$ variation)

| Method | $a$ | | | | |
|---|---|---|---|---|---|
| | 0.9 | 0.8 | 0.7 | 0.6 | 0.5 |
| ECD-aMCI | **0.875 ± 0.054** | **0.917 ± 0.037** | **0.928 ± 0.035** | **0.932 ± 0.034** | **0.931 ± 0.033** |
| PCMCI+ | 0.785 ± 0.066 | 0.857 ± 0.046 | 0.900 ± 0.037 | 0.916 ± 0.034 | 0.922 ± 0.033 |
| Bagged-PCMCI+ | 0.724 ± 0.069 | 0.779 ± 0.054 | 0.810 ± 0.048 | 0.808 ± 0.050 | 0.798 ± 0.050 |
| NTS-NOTEARS | 0.355 ± 0.122 | 0.564 ± 0.160 | 0.753 ± 0.059 | 0.850 ± 0.043 | 0.883 ± 0.042 |

Table 7: SHD Comparison ($a$ variation)

| Method | $a$ | | | | |
|---|---|---|---|---|---|
| | 0.9 | 0.8 | 0.7 | 0.6 | 0.5 |
| ECD-aMCI | **11.603 ± 4.102** | **8.903 ± 3.169** | **8.643 ± 3.107** | **9.410 ± 3.121** | **10.547 ± 3.126** |
| PCMCI+ | 16.673 ± 4.559 | 12.570 ± 3.393 | 10.560 ± 3.039 | 10.543 ± 2.958 | 11.207 ± 2.997 |
| Bagged-PCMCI+ | 22.203 ± 4.877 | 18.997 ± 4.030 | 17.947 ± 4.004 | 18.977 ± 4.247 | 20.717 ± 4.186 |
| NTS-NOTEARS | 67.903 ± 16.787 | 38.937 ± 10.174 | 22.957 ± 5.424 | 14.623 ± 4.004 | 11.793 ± 3.583 |

Table 8: Runtime ($s$) Comparison ($a$ variation)

| Method | $a$ | | | | |
|---|---|---|---|---|---|
| | 0.9 | 0.8 | 0.7 | 0.6 | 0.5 |
| ECD-aMCI | 35.68 ± 21.64 | 21.15 ± 12.93 | 18.80 ± 13.11 | 12.30 ± 4.22 | 11.53 ± 3.60 |
| PCMCI+ | 20.92 ± 13.96 | **10.80 ± 5.61** | **9.71 ± 6.63** | **6.66 ± 2.28** | **4.34 ± 0.97** |
| Bagged-PCMCI+ | 439.70 ± 120.31 | 359.30 ± 89.56 | 339.73 ± 89.41 | 330.18 ± 75.78 | 314.70 ± 74.63 |
| NTS-NOTEARS | **13.76 ± 9.98** | 15.59 ± 11.42 | 13.13 ± 5.02 | 14.15 ± 4.80 | 14.43 ± 5.52 |

Table 9: F1-score$_{\text{lagged}}$ Comparison (T variation)

| Method | T | | | |
|---|---|---|---|---|
| | 250 | 500 | 750 | 1000 |
| ECD-aMCI | **0.592 ± 0.134** | **0.848 ± 0.077** | **0.920 ± 0.055** | **0.945 ± 0.046** |
| PCMCI+ | 0.459 ± 0.135 | 0.695 ± 0.118 | 0.796 ± 0.106 | 0.844 ± 0.080 |
| Bagged-PCMCI+ | 0.374 ± 0.131 | 0.586 ± 0.121 | 0.705 ± 0.113 | 0.760 ± 0.093 |
| NTS-NOTEARS | 0.426 ± 0.191 | 0.493 ± 0.194 | 0.531 ± 0.158 | 0.537 ± 0.164 |

Table 10: F1-score$_{\text{all}}$ Comparison (T variation)

| Method | T | | | |
|---|---|---|---|---|
| | 250 | 500 | 750 | 1000 |
| ECD-aMCI | **0.781 ± 0.058** | **0.917 ± 0.037** | **0.954 ± 0.027** | **0.967 ± 0.024** |
| PCMCI+ | 0.709 ± 0.060 | 0.857 ± 0.046 | 0.909 ± 0.043 | 0.932 ± 0.032 |
| Bagged-PCMCI+ | 0.647 ± 0.062 | 0.779 ± 0.054 | 0.844 ± 0.051 | 0.874 ± 0.043 |
| NTS-NOTEARS | 0.522 ± 0.173 | 0.564 ± 0.160 | 0.607 ± 0.135 | 0.610 ± 0.142 |

Table 11: SHD Comparison (T variation)

| Method | T | | | |
|---|---|---|---|---|
| | 250 | 500 | 750 | 1000 |
| ECD-aMCI | **19.317 ± 3.854** | **8.903 ± 3.169** | **5.267 ± 2.631** | **3.783 ± 2.330** |
| PCMCI+ | 23.360 ± 3.556 | 12.570 ± 3.393 | 8.123 ± 3.421 | 5.857 ± 2.586 |
| Bagged-PCMCI+ | 28.003 ± 4.234 | 18.997 ± 4.030 | 14.283 ± 4.167 | 11.780 ± 3.603 |
| NTS-NOTEARS | 39.940 ± 7.682 | 38.937 ± 10.174 | 39.063 ± 7.767 | 37.303 ± 6.749 |

Table 12: Runtime ($s$) Comparison (T variation)

| Method | T | | | |
|---|---|---|---|---|
| | 250 | 500 | 750 | 1000 |
| ECD-aMCI | 9.790 ± 2.842 | 21.154 ± 12.929 | 16.803 ± 5.779 | 17.980 ± 7.912 |
| PCMCI+ | **3.375 ± 0.887** | **10.797 ± 5.614** | **8.548 ± 2.449** | **10.487 ± 3.330** |
| Bagged-PCMCI+ | 235.378 ± 51.948 | 359.303 ± 89.555 | 512.583 ± 113.982 | 589.065 ± 137.459 |
| NTS-NOTEARS | 10.321 ± 3.378 | 15.589 ± 11.417 | 22.436 ± 15.596 | 18.564 ± 4.539 |

Table 13: F1-score$_{\text{lagged}}$ Comparison (d variation)

| Method | d | | | |
|---|---|---|---|---|
| | 10 | 20 | 30 | 40 |
| ECD-aMCI | **0.945 ± 0.046** | **0.935 ± 0.031** | **0.917 ± 0.028** | **0.897 ± 0.027** |
| PCMCI+ | 0.844 ± 0.080 | 0.833 ± 0.101 | 0.807 ± 0.108 | 0.785 ± 0.115 |
| Bagged-PCMCI+ | 0.760 ± 0.093 | 0.739 ± 0.095 | 0.700 ± 0.095 | 0.669 ± 0.097 |
| NTS-NOTEARS | 0.537 ± 0.164 | 0.486 ± 0.176 | 0.423 ± 0.189 | 0.388 ± 0.198 |

Table 14: F1-score$_{\text{all}}$ Comparison (d variation)

| Method | d | | | |
|---|---|---|---|---|
| | 10 | 20 | 30 | 40 |
| ECD-aMCI | **0.967 ± 0.024** | **0.957 ± 0.018** | **0.944 ± 0.016** | **0.931 ± 0.018** |
| PCMCI+ | 0.932 ± 0.032 | 0.924 ± 0.042 | 0.909 ± 0.047 | 0.897 ± 0.050 |
| Bagged-PCMCI+ | 0.874 ± 0.043 | 0.839 ± 0.044 | 0.805 ± 0.042 | 0.776 ± 0.043 |
| NTS-NOTEARS | 0.610 ± 0.142 | 0.563 ± 0.161 | 0.510 ± 0.184 | 0.480 ± 0.189 |

Table 15: SHD Comparison (d variation)

| Method | d | | | |
|---|---|---|---|---|
| | 10 | 20 | 30 | 40 |
| ECD-aMCI | **3.783 ± 2.330** | **9.077 ± 3.445** | **15.963 ± 4.585** | **26.157 ± 7.028** |
| PCMCI+ | 5.857 ± 2.586 | 12.957 ± 6.062 | 21.823 ± 10.238 | 32.750 ± 14.379 |
| Bagged-PCMCI+ | 11.780 ± 3.603 | 29.827 ± 6.990 | 53.683 ± 9.811 | 82.400 ± 12.798 |
| NTS-NOTEARS | 37.303 ± 6.749 | 79.283 ± 11.669 | 126.643 ± 13.792 | 170.520 ± 17.492 |

Table 16: Runtime ($s$) Comparison (d variation)

| Method | d | | | |
|---|---|---|---|---|
| | 10 | 20 | 30 | 40 |
| ECD-aMCI | 17.980 ± 7.912 | 68.724 ± 18.949 | 160.456 ± 66.461 | 337.322 ± 183.383 |
| PCMCI+ | **10.487 ± 3.330** | **38.660 ± 13.031** | **85.480 ± 32.818** | **176.178 ± 102.563** |
| Bagged-PCMCI+ | 589.065 ± 137.459 | 2571.851 ± 699.392 | 6732.931 ± 2098.731 | 13243.336 ± 3738.661 |
| NTS-NOTEARS | 18.564 ± 4.539 | 99.162 ± 53.911 | 384.579 ± 188.730 | 795.130 ± 407.888 |

**Hyperparameters**  All hyperparameters required for the experiments in Figure 2 are provided in the following.

- **ECD-aMCI, PCMCI+:**  Confidence level $\alpha = 0.01$, maximum time lag $\tau_{\max} = 5$, condition independence test $CI = ParCorr$.

- **Bagged-PCMCI+:**  Confidence level $\alpha = 0.01$, maximum time lag $\tau_{\max} = 5$, condition independence test $CI = ParCorr$, boot samples $B_{\text{bagged}} = 50$.

- **NTS-NOTEARS:**  $\lambda_1 = 0.01$ for $T \in \{250, 500\}$, $\lambda_1 = 0.001$ for $T \in \{750, 1250\}$, $\lambda_2 = 0.05$, $K = 5$, $m = d$, the number of hidden layers = 1.

### A.5.2  DETAILED RESULTS AND HYPERPARAMETERS OF NONLINEAR SETTINGS (FIGURE 3)

This section presents the parameter settings and detailed results of simulation experiments under nonlinear settings. Subsequently, all hyperparameters required for the experiments in nonlinear settings are provided.

**Parameter setting:** The datasets in nonlinear settings are generated from the same model as in linear settings:

$$X_{j,t} = a_j X_{j,t-1} + \sum_{i=1}^{d} \sum_{\tau=1}^{\tau_{\max}} \delta_{i,j,\tau} c_{i,j,\tau} f_i(X_{i,t-\tau}) + \varepsilon_{j,t}, \tag{5}$$

The coefficient $c_{i,j,\tau}$, which represents the strength of the link $X_{i,t-\tau} \to X_{j,t}$, is drawn uniformly from $[0.15, 0.25]$. For each simulated dataset, $3 \cdot d$ cross-links between variables are randomly selected. 50% of the links are set to be contemporaneous ($\tau = 0$), and the remaining links have time lags $\tau$ uniformly drawn from $\{1, \ldots, \tau_{\text{true}}\}$, where true maximum time lag $\tau_{\text{true}}$ is set to 3. The functional dependencies $f_{i,j}(x) = \tanh(x)$.

**Results:** As shown in Figure 3, ECD-aMCI consistently outperforms the baseline algorithms across most evaluation metrics. In the first column, as the autocorrelation strength increases from 0.5 to 0.9, ECD-aMCI maintains superior F1-scores, with improvements of approximately 5-10% over competing algorithms, while keeping the SHD consistently lower. Although the performance advantage is modest at lower sample size, the second column reveals that the superiority of the ECD-aMCI algorithm becomes increasingly pronounced as the sample size grows. The third column illustrates that ECD-aMCI exhibits stable performance with increasing dimensionality, maintaining relatively consistent F1-scores even as the number of variables increases. The bottom row illustrates that constraint-based algorithms (including ECD-aMCI) require longer computational time than optimization-based algorithms in nonlinear settings, primarily due to the computational intensity of nonparametric conditional independence tests. Future research could explore more efficient conditional independence test to reduce computational runtime. These simulation results demonstrate that the ECD-aMCI algorithm offers substantial advantages over competing algorithms, particularly in settings with sufficient sample size. Note that Bagged-PCMCI+ is not included in the nonlinear setting comparisons due to its prohibitively high computational and memory requirements.

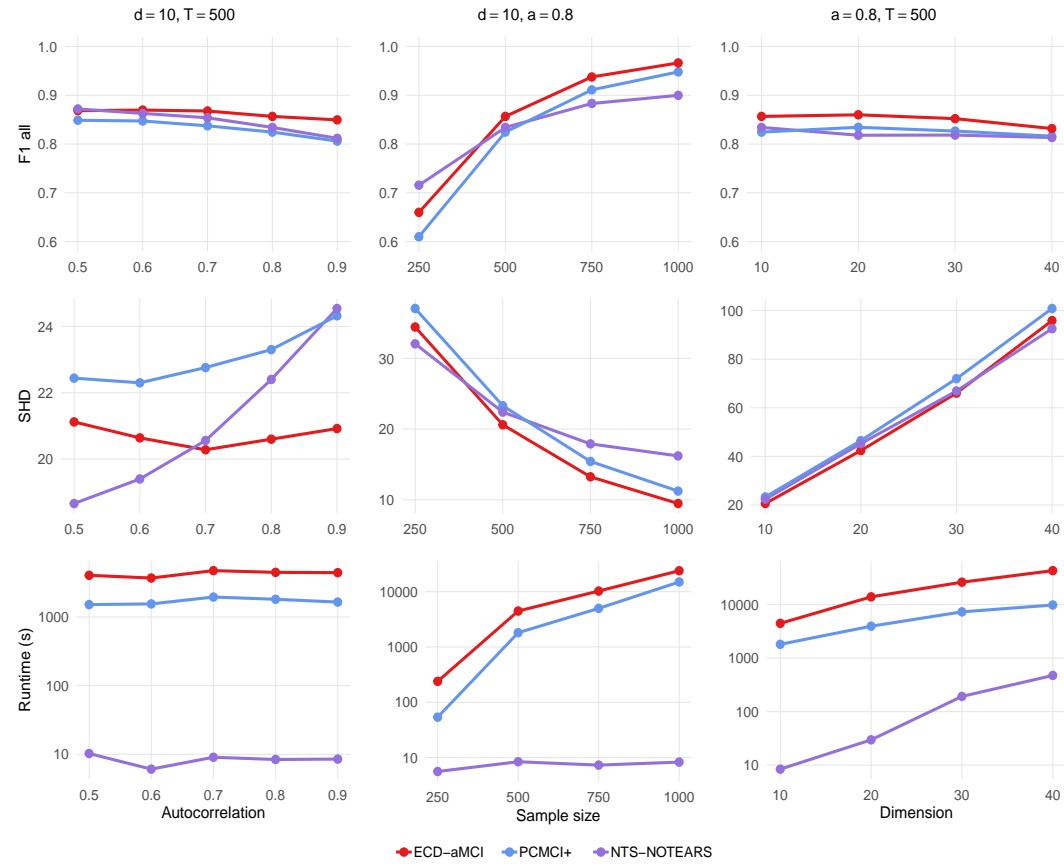

Figure 3: Mean metrics over 50 datasets for each nonlinear linear setting.

According to the standard errors presented in Tables 17-28, the ECD-aMCI algorithm demonstrates the smallest standard error for F1-score$_{\text{lagged}}$, F1-score$_{\text{all}}$ and SHD across most settings.

Table 17: F1-score$_{\text{lagged}}$ Comparison (auto variation)

| Method | $a$ | | | | |
|---|---|---|---|---|---|
| | 0.9 | 0.8 | 0.7 | 0.6 | 0.5 |
| ECD-aMCI | **0.737 ± 0.101** | **0.752 ± 0.107** | **0.778 ± 0.093** | **0.794 ± 0.079** | **0.816 ± 0.077** |
| PCMCI+ | 0.685 ± 0.117 | 0.713 ± 0.103 | 0.738 ± 0.101 | 0.763 ± 0.097 | 0.784 ± 0.092 |
| NTS-NOTEARS | 0.718 ± 0.103 | 0.745 ± 0.091 | 0.766 ± 0.084 | 0.789 ± 0.078 | 0.809 ± 0.081 |

Table 18: F1-score$_{\text{all}}$ Comparison ($a$ variation)

| Method | $a$ | | | | |
|---|---|---|---|---|---|
| | 0.9 | 0.8 | 0.7 | 0.6 | 0.5 |
| ECD-aMCI | **0.850 ± 0.037** | **0.857 ± 0.039** | **0.868 ± 0.036** | **0.870 ± 0.032** | 0.869 ± 0.035 |
| PCMCI+ | 0.806 ± 0.045 | 0.825 ± 0.040 | 0.837 ± 0.038 | 0.847 ± 0.034 | 0.849 ± 0.035 |
| NTS-NOTEARS | 0.812 ± 0.046 | 0.834 ± 0.041 | 0.854 ± 0.041 | 0.863 ± 0.039 | **0.872 ± 0.040** |

Table 19: SHD Comparison ($a$ variation)

| Method | $a$ | | | | |
|---|---|---|---|---|---|
| | 0.9 | 0.8 | 0.7 | 0.6 | 0.5 |
| ECD-aMCI | **20.920 ± 3.123** | **20.600 ± 3.280** | **20.280 ± 2.892** | 20.640 ± 2.544 | 21.120 ± 2.903 |
| PCMCI+ | 24.320 ± 3.552 | 23.300 ± 3.132 | 22.760 ± 3.178 | 22.300 ± 2.744 | 22.440 ± 2.815 |
| NTS-NOTEARS | 24.540 ± 4.679 | 22.400 ± 3.863 | 20.560 ± 4.253 | **19.400 ± 3.950** | **18.660 ± 4.246** |

Table 20: Runtime ($s$) Comparison ($a$ variation)

| Method | $a$ | | | | |
|---|---|---|---|---|---|
| | 0.9 | 0.8 | 0.7 | 0.6 | 0.5 |
| ECD-aMCI | 4423.80 ± 696.92 | 4467.35 ± 876.32 | 4733.17 ± 823.10 | 3692.25 ± 836.55 | 4046.83 ± 1067.61 |
| PCMCI+ | 1649.92 ± 388.47 | 1811.52 ± 414.25 | 1951.26 ± 465.31 | 1550.20 ± 356.56 | 1515.32 ± 424.96 |
| NTS-NOTEARS | **8.52 ± 1.05** | **8.40 ± 1.41** | **9.06 ± 1.26** | **6.08 ± 0.51** | **10.27 ± 2.06** |

Table 21: F1-score$_{\text{lagged}}$ Comparison (T variation)

| Method | T | | | |
|---|---|---|---|---|
| | 250 | 500 | 750 | 1000 |
| ECD-aMCI | 0.425 ± 0.141 | **0.752 ± 0.107** | **0.890 ± 0.063** | **0.944 ± 0.045** |
| PCMCI+ | 0.387 ± 0.137 | 0.713 ± 0.103 | 0.839 ± 0.083 | 0.899 ± 0.046 |
| NTS-NOTEARS | **0.566 ± 0.094** | 0.745 ± 0.091 | 0.824 ± 0.074 | 0.841 ± 0.060 |

Table 22: F1-score$_{\text{all}}$ Comparison (T variation)

| Method | T | | | |
|---|---|---|---|---|
| | 250 | 500 | 750 | 1000 |
| ECD-aMCI | 0.660 ± 0.052 | **0.857 ± 0.039** | **0.938 ± 0.027** | **0.966 ± 0.022** |
| PCMCI+ | 0.610 ± 0.051 | 0.825 ± 0.040 | 0.911 ± 0.030 | 0.948 ± 0.024 |
| NTS-NOTEARS | **0.716 ± 0.051** | 0.834 ± 0.041 | 0.883 ± 0.034 | 0.900 ± 0.031 |

Table 23: SHD Comparison (T variation)

| Method | T | | | |
|---|---|---|---|---|
| | 250 | 500 | 750 | 1000 |
| ECD-aMCI | $34.440 \pm 3.226$ | $\mathbf{20.600 \pm 3.280}$ | $\mathbf{13.240 \pm 3.734}$ | $\mathbf{9.460 \pm 3.390}$ |
| PCMCI+ | $37.060 \pm 2.760$ | $23.300 \pm 3.132$ | $15.420 \pm 3.909$ | $11.220 \pm 3.324$ |
| NTS-NOTEARS | $\mathbf{32.060 \pm 4.688}$ | $22.400 \pm 3.863$ | $17.900 \pm 4.239$ | $16.200 \pm 3.980$ |

Table 24: Runtime ($s$) Comparison (T variation)

| Method | T | | | |
|---|---|---|---|---|
| | 250 | 500 | 750 | 1000 |
| ECD-aMCI | $238.83 \pm 31.69$ | $4467.35 \pm 876.32$ | $10259.70 \pm 2007.56$ | $23890.22 \pm 3613.27$ |
| PCMCI+ | $53.76 \pm 9.85$ | $1811.52 \pm 414.25$ | $4985.80 \pm 1353.46$ | $14973.02 \pm 3799.10$ |
| NTS-NOTEARS | $\mathbf{5.60 \pm 0.35}$ | $\mathbf{8.40 \pm 1.41}$ | $\mathbf{7.31 \pm 0.75}$ | $\mathbf{8.28 \pm 0.74}$ |

Table 25: F1-score$_{\text{lagged}}$ Comparison (d variation)

| Method | d | | | |
|---|---|---|---|---|
| | 10 | 20 | 30 | 40 |
| ECD-aMCI | $\mathbf{0.752 \pm 0.107}$ | $\mathbf{0.782 \pm 0.057}$ | $\mathbf{0.755 \pm 0.043}$ | $\mathbf{0.733 \pm 0.047}$ |
| PCMCI+ | $0.713 \pm 0.103$ | $0.757 \pm 0.068$ | $0.731 \pm 0.051$ | $0.730 \pm 0.048$ |
| NTS-NOTEARS | $0.745 \pm 0.091$ | $0.741 \pm 0.072$ | $0.727 \pm 0.046$ | $0.729 \pm 0.050$ |

Table 26: F1-score$_{\text{all}}$ Comparison (d variation)

| Method | d | | | |
|---|---|---|---|---|
| | 10 | 20 | 30 | 40 |
| ECD-aMCI | $\mathbf{0.857 \pm 0.039}$ | $\mathbf{0.860 \pm 0.026}$ | $\mathbf{0.852 \pm 0.021}$ | $\mathbf{0.832 \pm 0.024}$ |
| PCMCI+ | $0.825 \pm 0.040$ | $0.834 \pm 0.030$ | $0.827 \pm 0.021$ | $0.816 \pm 0.023$ |
| NTS-NOTEARS | $0.834 \pm 0.041$ | $0.818 \pm 0.036$ | $0.818 \pm 0.020$ | $0.813 \pm 0.026$ |

Table 27: SHD Comparison (d variation)

| Method | d | | | |
|---|---|---|---|---|
| | 10 | 20 | 30 | 40 |
| ECD-aMCI | $\mathbf{20.600 \pm 3.280}$ | $\mathbf{42.400 \pm 5.223}$ | $\mathbf{66.040 \pm 6.579}$ | $95.880 \pm 9.024$ |
| PCMCI+ | $23.300 \pm 3.132$ | $46.500 \pm 5.442$ | $71.980 \pm 5.941$ | $100.840 \pm 7.630$ |
| NTS-NOTEARS | $22.400 \pm 3.863$ | $45.240 \pm 6.445$ | $66.900 \pm 6.655$ | $\mathbf{92.540 \pm 9.104}$ |

Table 28: Runtime ($s$) Comparison (d variation)

| Method | d | | | |
|---|---|---|---|---|
| | 10 | 20 | 30 | 40 |
| ECD-aMCI | $4467.35 \pm 876.32$ | $13977.40 \pm 3313.75$ | $26279.66 \pm 4543.04$ | $43190.11 \pm 8954.29$ |
| PCMCI+ | $1811.52 \pm 414.25$ | $3949.59 \pm 1480.62$ | $7322.64 \pm 1707.84$ | $9831.77 \pm 2665.12$ |
| NTS-NOTEARS | $\mathbf{8.40 \pm 1.41}$ | $\mathbf{29.66 \pm 3.47}$ | $\mathbf{192.25 \pm 84.32}$ | $\mathbf{473.98 \pm 173.88}$ |

**Hyperparameters:** All hyperparameters required for the experiments in Figure 1 are provided in the following.

- **ECD-aMCI, PCMCI+:** Confidence level $\alpha = 0.01$, maximum time lag $\tau_{\max} = 3$, condition independence test $CI = GPDC$.

- **NTS-NOTEARS:** $\lambda_1 = 0.01$ for $T \in \{250, 500\}$, $\lambda_1 = 0.001$ for $T \in \{750, 1250\}$, $\lambda_2 = 0.05$, $K = 3$, $m = d$, the number of hidden layers = 1.

## B THE USE OF LARGE LANGUAGE MODELS

Large language models were used in polishing the paper by correcting grammar and spelling errors.

