# OpenReview forum: "Enhanced Causal Discovery for Autocorrelated Time Series via Adaptive Momentary Conditional Independence"
_ICLR.cc/2026/Conference — ICLR 2026 Conference Withdrawn Submission_

### Official Review · Reviewer_kPPS · 2025-10-29

**Soundness:** 2
**Presentation:** 4
**Contribution:** 3
**Rating:** 6
**Confidence:** 3

**Summary:**

This paper aims at two challenges in time-series causal discovery: i) autocorrelation, and ii) contemporaneous effect. The authors first propose aMCI to enhance signal-to-noise ratio and improve detection power without inflating false positives, and then leverages a three-phase ECD-aMCI algorithm. Experiments are conducted on both synthetic and fMRI benchmark data

**Strengths:**

S1: The paper is technically rigorous and empirically thorough. The authors prove consistency, order independence, and enhanced accuracy under standard assumptions. And the robust anlysis and runtime analysis is conducted.

S2: The paper is well-written. The motivating example (Section 3.1) clearly illustrates why standard MCI fails and how aMCI helps, using intuitive signal-to-noise reasoning backed by algebraic decomposition.

**Weaknesses:**

W1: Incomplete coverage of related work on autocorrelation-aware causal discovery. The paper overlooks several recent methods that explicitly address autocorrelation in time-series causal discovery, such as **CDANS** (Temporal causal discovery from autocorrelated and non-stationary time series data) mentioned in the survey. The absence of a discussion (in Section 1 or 2) or empirical comparison with such methods weakens the contextualization of the proposed contribution. I would like to reconsider the contribution score if this gap can be addressed.

W2: Experimental evaluation lacks real-world validation beyond fMRI. The paper claims relevance to "earth science, neuroscience, and economics" at the very beginning, but no experiments on real-world time series (e.g., climate data, financial markets, ICU patient monitoring). The soundness has only been witnessed in one domain (i.e., fMRI). I would like to reconsider the soundness score if this issue can be solved.

**Questions:**

See the weakness above.

---

> ### Author Response · Authors · 2025-11-22
> **Response to Reviewer kPPS**
>
> We sincerely thank the reviewer for the insightful comments.
>
> > **W1:**  The paper overlooks several recent methods that explicitly address autocorrelation in time-series causal discovery, such as **CDANS**  (Temporal causal discovery from autocorrelated and non-stationary time series data) mentioned in the survey.
>
> **R1:**  As a recent temporal causal discovery approach that considers both autocorrelation and non-stationarity properties of time series, the CDANS method [1] fully inherits the aMCI approach from the PCMCI+ algorithm for addressing autocorrelation. Since CDANS adopts the same aMCI mechanism as PCMCI+ for handling autocorrelation, which is the focus of our work, we limit our empirical comparisons to PCMCI+ to avoid redundancy in evaluating this specific aspect. Nevertheless, to ensure completeness in our related work section, we will add a discussion of CDANS in Section 1, acknowledging its contributions to handling non-stationary time series data.
>
>
>
> > **W2:** Experimental evaluation lacks real-world validation beyond fMRI.
>
> **R2:**  We have now included additional real-world validation on telecommunication network alarm data [2]. In telecommunication networks, a single fault can trigger a cascade of various alarm types across multiple connected devices. Understanding the causal relationships among these alarms is crucial for intelligent alarm management and fault root cause analysis, as manually handling such alarm floods can quickly overwhelm operators.
>
> We conducted an experimental evaluation on observable historical alarm data (time series) from a real-world telecommunication network. We applied the proposed algorithm and baseline methods to learn causal alarm graphs, where each node represents a distinct alarm type. The experimental results in Table 1 demonstrate that our method remains competitive in this practical application, further validating its effectiveness beyond the fMRI domain.
>
> **Table 1:  Evaluation on observable historical alarm data**
>
> |               | F1-score$_\text{all}$ | SHD     |
> | ------------- | --------------------- | ------- |
> | ECD-aMCI      | **0.5872**            | **103** |
> | PCMCI+        | 0.5524                | 109     |
> | Bagged-PCMCI+ | 0.5000                | 120     |
> | NTS-NOTEARS   | 0.3956                | 110     |
>
>
>
> **Reference**
>
> [1] Ferdous, M. H., Hasan, U., & Gani, M. O. (2023, December). Cdans: Temporal causal discovery from autocorrelated and non-stationary time series data. In *Machine Learning for Healthcare Conference* (pp. 186-207). PMLR.
>
> [2] https://competition.huaweicloud.com/information/1000041487/dataset

---

> > ### Author Response · Authors · 2025-11-27
> > **Additional notes on W1**
> >
> > > W1: Incomplete coverage of related work on autocorrelation-aware causal discovery. The absence of a discussion (in Section 1 or 2) or empirical comparison with such methods weakens the contextualization of the proposed contribution.
> >
> > As detailed in Appendix A.2, we selected our baselines from a pool of ten algorithms covering the four major families of causal discovery methods for handling autocorrelation. We filtered these methods based on whether they learn a window causal graph and whether they handle contemporaneous effects, which resulted in five candidates (PCMCI+, Bagged-PCMCI+, NTS-NOTEARS, DYNOTEARS, and VAR-LiNGAM). Following prior literature, we then selected the three state-of-the-art methods among them as our final baselines. Table 4 in the paper provides a  summary of the rationale behind this selection process. Following your suggestion, we will move this discussion to the introduction section for better visibility.
> >
> > To address your concerns more comprehensively, we have now included DYNOTEARS (score-based) and VAR-LiNGAM (structural causal model-based) as additional baseline methods in our evaluation. The corresponding experimental results are provided below.
> >
> > **Table 1: F1-score$_{\text{all}}$ Comparison ($a$ variation)**
> > |Method|a=0.9|a=0.8|a=0.7|a=0.6|a=0.5|
> > |---|---|---|---|---|---|
> > |ECD-aMCI|**0.87±0.05**|**0.92±0.04**|**0.93±0.04**|**0.93±0.03**|**0.93±0.03**|
> > |PCMCI+|0.78±0.07|0.86±0.05|0.90±0.04|0.92±0.03|0.92±0.03|
> > |Bagged-PCMCI+|0.72±0.07|0.78±0.05|0.81±0.05|0.81±0.05|0.80±0.05|
> > |NTS-NOTEARS|0.35±0.12|0.56±0.16|0.75±0.06|0.85±0.04|0.88±0.04|
> > |VAR-LiNGAM|0.52±0.17|0.74±0.15|0.86±0.05|0.89±0.04|0.89±0.04|
> > |DYNOTEARS|0.27±0.07|0.23±0.08|0.17±0.08|0.12±0.07|0.08±0.06|
> >
> > **Table 2: F1-lagged Comparison ($a$ variation)**
> > |Method|a=0.9|a=0.8|a=0.7|a=0.6|a=0.5|
> > |---|---|---|---|---|---|
> > |ECD-aMCI|**0.78±0.11**|**0.85±0.08**|**0.87±0.07**|**0.88±0.07**|**0.89±0.06**|
> > |PCMCI+|0.52±0.16|0.69±0.12|0.81±0.09|0.85±0.07|0.87±0.07|
> > |Bagged-PCMCI+|0.46±0.15|0.60±0.12|0.67±0.10|0.68±0.10|0.67±0.09|
> > |NTS-NOTEARS|0.24±0.15|0.49±0.19|0.71±0.09|0.80±0.07|0.84±0.07|
> > |VAR-LiNGAM|0.03±0.06|0.04±0.06|0.03±0.06|0.03±0.06|0.03±0.06|
> > |DYNOTEARS|0.29±0.20|0.60±0.21|0.77±0.08|0.81±0.07|0.81±0.07|
> >
> > **Table 3: SHD Comparison ($a$ variation)**
> > |Method|a=0.9|a=0.8|a=0.7|a=0.6|a=0.5|
> > |---|---|---|---|---|---|
> > |ECD-aMCI|**11.60±4.10**|**8.90±3.17**|**8.64±3.11**|**9.41±3.12**|10.55±3.13|
> > |PCMCI+|16.67±4.56|12.57±3.39|10.56±3.04|10.54±2.96|11.21±3.00|
> > |Bagged-PCMCI+|22.36±5.08|18.81±3.89|18.01±4.01|19.24±4.46|20.48±4.33|
> > |NTS-NOTEARS|67.85±16.69|40.21±11.27|23.32±5.58|14.71±4.07|11.69±3.59|
> > |VAR-LiNGAM|41.62±2.42|42.57±2.48|43.53±2.23|44.33±1.89|45.14±1.99|
> > |DYNOTEARS|37.64±11.90|22.17±12.17|12.57±4.44|10.47±3.63|**10.37±3.53**|

---

### Official Review · Reviewer_xj8c · 2025-10-31

**Soundness:** 2
**Presentation:** 3
**Contribution:** 2
**Rating:** 4
**Confidence:** 3

**Summary:**

This paper addresses causal discovery in autocorrelated time series by introducing the Adaptive Momentary Conditional Independence method and the Enhanced Causal Discovery via aMCI algorithm.

**Strengths:**

The key insight is that strong autocorrelation can mask causal relationships when conditioning on immediate predecessors. The aMCI method adaptively modifies conditioning sets by replacing immediate predecessors with earlier lag variables when they are not confounders. The ECD-aMCI algorithm employs a three-phase approach: initial lagged parent estimation, refinement via aMCI, and complete skeleton discovery. Experiments on simulated and benchmark datasets demonstrate improved detection of lagged links compared to baseline methods.

**Weaknesses:**

1: While the method shows consistent improvements, the gains are often modest (5-15% in many cases). For high autocorrelation settings where the method should excel most, the absolute performance remains relatively low.

2: The aMCI method requires multiple conditional independence tests per edge (up to 3 for contemporaneous links), significantly increasing computational cost compared to standard methods, especially for nonlinear settings.

3: The method relies on correctly identifying confounders vs. non-confounders, which requires accurate initial parent set estimation. In practice, this distinction may be unreliable, potentially limiting the method's effectiveness.

4: The paper primarily compares against PCMCI+ variants and one optimization-based method (NTS-NOTEARS). Missing comparisons with other recent time series causal discovery methods and specifically methods designed for handling autocorrelation.

5: While claimed to be hyperparameter-insensitive, the method still requires setting maximum lag τmax and confidence level α. The robustness analysis for τmax shows performance degradation as this parameter increases.

6: The three-phase approach adds complexity, and the adaptive conditioning strategy may be difficult to interpret and debug in practice compared to standard constraint-based methods.

**Questions:**

The work makes a reasonable contribution to understanding how to better handle autocorrelation in causal discovery, but the impact is somewhat incremental given the additional complexity introduced.

---

> ### Author Response · Authors · 2025-11-22
> **Response to Reviewer xj8c (1/2)**
>
> We sincerely thank you for the insightful comments.
>
> > **W1:** While the method shows consistent improvements, the gains are often modest (5-15% in many cases). For high autocorrelation settings where the method should excel most, the absolute performance remains relatively low.
>
>  **R1:** For high-autocorrelation settings such as $a = 0.9$, ECD-aMCI achieves an F1 score of 87.5%, indicating that the method remains effective. This performance compares favorably with established methods designed for handling autocorrelation, such as PCMCI, PCMCI+ which typically report lower absolute performance in similar high-autocorrelation scenarios.
>
> > **W2:** The aMCI method requires multiple conditional independence tests per edge (up to 3 for contemporaneous links), significantly increasing computational cost compared to standard methods, especially for nonlinear settings.
>
> **R2:** As you correctly pointed out, in the worst case the aMCI method requires at most two additional CI tests compared to PCMCI+, which does introduce some extra computational cost. However, the total runtime is at most three times that of PCMCI+, as confirmed by our simulation results, and therefore both methods remain within the same order of magnitude in practice. This relationship also holds under nonlinear settings; the runtime of both algorithms increases, but this is primarily due to the inherently higher computational cost of nonlinear CI tests rather than the design of aMCI method.
>
> > **W3:** The method relies on correctly identifying confounders vs. non-confounders, which requires accurate initial parent set estimation. In practice, this distinction may be unreliable, potentially limiting the method's effectiveness.
>
> **R3:** As shown in Step 1 of the Consistency Proof in Appendix A.4.2, the estimated lagged parent set is a superset of the true parent set. This implies that our method may label more nodes as potential confounders than those that actually are. Since aMCI only replaces nodes that are identified as non-confounders, the number of replacements it performs is smaller than the number of nodes that would need replacement in the ideal case. Therefore, our modification of the conditioning set is conservative, which provides robustness and tolerance to the type of issue you mentioned.
>
> > **W4:** The paper primarily compares against PCMCI+ variants and one optimization-based method (NTS-NOTEARS). Missing comparisons with other recent time series causal discovery methods and specifically methods designed for handling autocorrelation.
>
> **R4:**  In fact, we selected our baselines from a pool of ten algorithms covering the four major families of causal discovery methods for handling autocorrelation. We filtered these methods based on whether they learn a window causal graph and whether they handle contemporaneous effects, which resulted in five candidates (PCMCI+, Bagged-PCMCI+, NTS-NOTEARS, DYNOTEARS, and VAR-LiNGAM). Following prior literature, we then selected the three state-of-the-art methods among them as our final baselines. As detailed in Appendix A.2 and Table 4, we have already explained the rationale behind choosing these three baselines.
>
> To address your concerns more comprehensively, we have now included DYNOTEARS (score-based) and VAR-LiNGAM (structural causal model-based) as additional baseline methods in our evaluation. The corresponding experimental results are provided below.
>
> **Table 1: F1-score$_{\text{all}}$ Comparison ($a$ variation)**
> |Method|a=0.9|a=0.8|a=0.7|a=0.6|a=0.5|
> |---|---|---|---|---|---|
> |ECD-aMCI|**0.87±0.05**|**0.92±0.04**|**0.93±0.04**|**0.93±0.03**|**0.93±0.03**|
> |PCMCI+|0.78±0.07|0.86±0.05|0.90±0.04|0.92±0.03|0.92±0.03|
> |Bagged-PCMCI+|0.72±0.07|0.78±0.05|0.81±0.05|0.81±0.05|0.80±0.05|
> |NTS-NOTEARS|0.35±0.12|0.56±0.16|0.75±0.06|0.85±0.04|0.88±0.04|
> |VAR-LiNGAM|0.52±0.17|0.74±0.15|0.86±0.05|0.89±0.04|0.89±0.04|
> |DYNOTEARS|0.27±0.07|0.23±0.08|0.17±0.08|0.12±0.07|0.08±0.06|
>
> **Table 2: F1-lagged Comparison ($a$ variation)**
> |Method|a=0.9|a=0.8|a=0.7|a=0.6|a=0.5|
> |---|---|---|---|---|---|
> |ECD-aMCI|**0.78±0.11**|**0.85±0.08**|**0.87±0.07**|**0.88±0.07**|**0.89±0.06**|
> |PCMCI+|0.52±0.16|0.69±0.12|0.81±0.09|0.85±0.07|0.87±0.07|
> |Bagged-PCMCI+|0.46±0.15|0.60±0.12|0.67±0.10|0.68±0.10|0.67±0.09|
> |NTS-NOTEARS|0.24±0.15|0.49±0.19|0.71±0.09|0.80±0.07|0.84±0.07|
> |VAR-LiNGAM|0.03±0.06|0.04±0.06|0.03±0.06|0.03±0.06|0.03±0.06|
> |DYNOTEARS|0.29±0.20|0.60±0.21|0.77±0.08|0.81±0.07|0.81±0.07|
>
> **Table 3: SHD Comparison ($a$ variation)**
> |Method|a=0.9|a=0.8|a=0.7|a=0.6|a=0.5|
> |---|---|---|---|---|---|
> |ECD-aMCI|**11.60±4.10**|**8.90±3.17**|**8.64±3.11**|**9.41±3.12**|10.55±3.13|
> |PCMCI+|16.67±4.56|12.57±3.39|10.56±3.04|10.54±2.96|11.21±3.00|
> |Bagged-PCMCI+|22.36±5.08|18.81±3.89|18.01±4.01|19.24±4.46|20.48±4.33|
> |NTS-NOTEARS|67.85±16.69|40.21±11.27|23.32±5.58|14.71±4.07|11.69±3.59|
> |VAR-LiNGAM|41.62±2.42|42.57±2.48|43.53±2.23|44.33±1.89|45.14±1.99|
> |DYNOTEARS|37.64±11.90|22.17±12.17|12.57±4.44|10.47±3.63|**10.37±3.53**|

---

> ### Author Response · Authors · 2025-11-22
> **Response to Reviewer xj8c (2/2)**
>
> > **W5:** While claimed to be hyperparameter-insensitive, the method still requires setting maximum lag $\tau_{\text{max}}$ and confidence level $\alpha$. The robustness analysis for τmax shows performance degradation as this parameter increases.
>
> **R5:** We claim that our method is hyperparameter-insensitive based on the following reasons:
>
> * **Minimal hyperparameter requirement**: Our method requires only 2 hyperparameters, the same as PCMCI+ but fewer than PCMCI+Bagged (3 hyperparameters) and NTS-NOTEARS (6 hyperparameters).
>
> * **Easy-to-determine hyperparameters**: Our hyperparameters are straightforward to determine. The significance level $\alpha$ is a standard hyperparameter in statistics with clear statistical meaning and common choices (0.01 or 0.05); $\tau_{\max}$ can be preliminarily determined by estimating the autocorrelation coefficient.
>
> * **Robustness to hyperparameter variations**: The algorithm exhibits low sensitivity to hyperparameter changes. When increasing $\tau_{\max}$ from 5 to 10 (doubling the value), the F1 score decreases by only 2%. Throughout all simulations in this paper, we consistently use the common significance level $\alpha = 0.01$. Our simulations encompass 26 different parameter settings: 5 levels of autocorrelation coefficient $a$, 4 settings for sample size $T$, 4 settings for dimension $d$, and 2 types of causal relationships (linear and nonlinear). For each configuration, we randomly generated 300 time series, each with randomly generated window causal graphs and causal strengths. The ability to achieve competitive performance using the same significance level across a total of 7,800 diverse time series datasets ($26 \times 300 = 7{,}800$) demonstrates that ECD-aMCI is hyperparameter-insensitive.
>
>
>
> > **W6:** The three-phase approach adds complexity, and the adaptive conditioning strategy may be difficult to interpret and debug in practice compared to standard constraint-based methods.
>
> **R6:** We appreciate this concern and would like to clarify both the interpretability and practical reliability of our approach:
>
> * **Interpretability**: We have provided justification for the adaptive conditioning strategy from two complementary perspectives: (1) through concrete examples in Section 3.1 that illustrate the mechanism, and (2) through signal-to-noise ratio (SNR) analysis that provides theoretical intuition for why the strategy improves detection power.
>
> * **Practical reliability**: Our extensive simulations across 7,800 diverse time series datasets with 26 different parameter configurations have consistently demonstrated robust performance, with no instances of algorithmic failures or unexpected behaviors.
>
> * **Complexity**: While the three-phase approach adds structure, each phase has a clear purpose and the conditioning set selection follows explicit rules based on graph properties, making the algorithm straightforward to implement and verify.

---

### Official Review · Reviewer_Avyq · 2025-11-01

**Soundness:** 3
**Presentation:** 3
**Contribution:** 2
**Rating:** 4
**Confidence:** 5

**Summary:**

This paper introduces Adaptive Momentary Conditional Independence (aMCI), a refinement of constraint-based causal discovery for autocorrelated time series that mitigates the masking of true causal links by strong temporal self-dependence. The core idea is that conditioning on highly autocorrelated recent lags can suppress genuine causal signals; thus, aMCI adaptively substitutes such immediate predecessors with earlier lags to improve detectability. Integrated into the PCMCI+ framework, this yields ECD-aMCI, an algorithm that retains PCMCI+’s structure but modifies the conditioning step to preserve causal signal strength. The authors provide theoretical guarantees for consistency, order independence, and enhanced accuracy, and demonstrate empirical gains over PCMCI+, Bagged-PCMCI+, and NTS-NOTEARS on linear, nonlinear, and fMRI datasets, particularly in high-autocorrelation regimes. While the approach is technically sound, clear, and effective, its algorithmic novelty is limited, as the overall structure, orientation rules, and independence tests remain unchanged from PCMCI+.

**Strengths:**

- The recognition that standard conditional independence testing can "over-condition" and suppress causal signals in highly autocorrelated series is a non-trivial and significant contribution. The paper clearly explains this phenomenon and proposes an intuitive solution.
- The authors provide important theoretical guarantees for their ECD-aMCI algorithm, including consistency, order independence, and a proof of "enhanced accuracy", which formalizes the intuitive benefit of the aMCI method.
- Across simulations and the fMRI benchmark, ECD-aMCI consistently improves the F1-score for lagged edges, especially under high autocorrelation, while maintaining reasonable computational cost.
- The paper is very well-motivated. The simple bivariate example in Section 3.1 provides a clear illustration of the exact problem the paper aims to solve.

**Weaknesses:**

- The ECD-aMCI algorithm closely follows the PCMCI+ pipeline in both design and implementation. The independence tests, search phases, and orientation rules are entirely retained, and the only substantive modification is the adaptive adjustment of the conditioning set before each conditional independence test. While this modification is intuitively appealing and theoretically supported, it represents a localized heuristic improvement rather than a new causal discovery framework. Consequently, the contribution resides primarily at the implementation level within the PCMCI+ structure rather than at the algorithmic or theoretical level.
- The evaluation on both stationary and non-stationary datasets is presented in a somewhat inconsistent and incomplete manner. While the authors include additional experiments on non-stationary data in Appendix C.3, these results are not integrated into the main text and are reported briefly, with limited quantitative interpretation. Moreover, the main body of the paper explicitly states that “only stationary models are considered” in the simulated data generation section, which is misleading; it is unclear whether this statement refers to the dataset itself or to the earlier stationarity assumption of the causal model. If it refers to the assumption, it should not appear in the data-generation description; if it refers to the dataset, then the inclusion of non-stationary experiments in the appendix contradicts that claim.
- The synthetic evaluation is also suboptimal in several respects. The nonlinear dataset generation described in the paper includes nonlinear transformations in equation form but omits realistic temporal characteristics such as trend or seasonality, which are ubiquitous in real-world time series. When visualized, the generated “nonlinear” time series exhibit no discernible non-stationary behavior or long-term structural variation, appearing qualitatively similar to linear autoregressive data. As a result, the nonlinear experiments test only pointwise nonlinearity rather than temporal non-stationarity, limiting the conclusions about robustness under realistic dynamics.
- The current choice of baselines (PCMCI+, Bagged-PCMCI+, and NTS-NOTEARS) does not sufficiently represent the state of the art in time-series causal discovery. The inclusion of Bagged-PCMCI+ is redundant, as it is simply a resampled ensemble variant of PCMCI+ that does not introduce a distinct methodological perspective. Instead, the evaluation should have incorporated modern approaches that provide complementary algorithmic principles or neural Granger-like approaches. Furthermore, at least one Granger-based causal discovery baseline, such as Neural Granger Causality (Tank et al., 2021 ) (https://ieeexplore.ieee.org/iel7/34/9812931/09376668.pdf ) or SPACETIME (Mameche et al., 2025)(https://ojs.aaai.org/index.php/AAAI/article/download/34136/36291 ) should have been included for completeness. Although Granger causality does not represent true structural causality under intervention semantics, it remains increasingly popular in applied time-series domains (finance, neuroscience, climate modeling) and provides a practical reference for temporal dependency detection. The absence of such baselines narrows the comparative context and underrepresents how ECD-aMCI performs relative to recent state-of-the-art causal discovery methods.
- The theoretical component of the paper largely extends existing PCMCI+ results to the adaptive setting. The proofs for consistency and order independence closely parallel those of the original framework, while the “enhanced accuracy” theorem formalizes an intuitively expected improvement in test power rather than establishing a new theoretical guarantee. There is no formal quantification of how much adaptive conditioning improves causal recovery beyond qualitative reasoning about signal-to-noise ratios. As a result, the theoretical novelty is incremental, and the contribution remains primarily empirical.

**Questions:**

1. Could a scaled-down Bagged-PCMCI+ comparison be included for nonlinear/fMRI data to complete the baseline picture?
2. How robust is aMCI to non-stationary processes (e.g., seasonality, drift)?
3. How does ECD-aMCI compare to nonlinear Granger models (e.g., NCG or RNN-based GC) in detecting lagged vs. contemporaneous dependencies?

---

> ### Author Response · Authors · 2025-11-22
> **Response to Reviewer Avyq (1/3)**
>
> We are truly grateful for your thorough review and the constructive comments, which have been very helpful for improving our work.
>
> > **W1:** While this modification is intuitively appealing and theoretically supported, it represents a localized heuristic improvement rather than a new causal discovery framework.
>
> **R1:**
>
> * **Specific contribution to time series causal discovery**: We first analyze how autocorrelation affects causal discovery and identify that standard conditioning sets can "over-condition" and suppress causal signals in highly autocorrelated series. This analysis provides guidance for improving all constraint-based causal discovery algorithms for tome series.
> * **Broader theoretical perspective**: Beyond the proposed algorithm, this insight offers a general improvement strategy for all constraint-based causal discovery algorithms (not limited to time series): theoretically equivalent design choices under ideal conditions may yield different practical performance, necessitating a preference for designs better suited to the data characteristics.
>
>
>
> > **W2: (1)** While the authors include additional experiments on non-stationary data in Appendix C.3, these results are not integrated into the main text and are reported briefly, with limited quantitative interpretation. **(2)** Moreover, the main body of the paper explicitly states that “only stationary models are considered” in the simulated data generation section, which is misleading; it is unclear whether this statement refers to the dataset itself or to the earlier stationarity assumption of the causal model.
>
> **R2:** **(1)** We would like to clarify what appears to be a misunderstanding. After carefully rechecking our manuscript, we confirm that there is no Appendix C.3 in our paper, and no non-stationary data were included in any of our experiments. We only provide additional experiments under nonlinear causal relationships in Appendix A.5.2, where the synthetic datasets are also stationary. This is consistent with your observation in **W3** that "when visualized, the generated nonlinear time series exhibit no discernible non-stationary behavior or long-term structural variation."
>
> **(2)** The phrase “only stationary models are considered” simply means that all generated datasets are stationary.
>
>
>
>
>
> > **W3 & Q2:** How robust is aMCI to non-stationary processes (e.g., seasonality, drift)? When visualized, the generated nonlinear time series exhibit no discernible non-stationary behavior or long-term structural variation. The nonlinear experiments test only pointwise nonlinearity rather than temporal non-stationarity, limiting the conclusions about robustness under realistic dynamics.
>
> **R3:** Our focus on stationary time series is motivated by two key considerations. First, even for stationary time series, causal discovery remains an incompletely solved problem. Second, algorithms designed for non-stationary time series are typically built upon foundations established for stationary cases.
>
> * Our work focuses on analyzing how strong autocorrelation affects the performance of causal discovery algorithms and, based on this analysis, proposes an improved algorithm, ECD-aMCI algorithm.
>
> * Regarding the SPACETIME algorithm [1] mentioned in your comment **W4**, while it is designed for non-stationary data, its core strategy involves identifying change points in the distribution and assuming stationarity between consecutive change points. Therefore, our insights can potentially transfer to algorithms for non-stationary time series. We recognize this as a promising direction for future work and will incorporate this limitation and future research direction into the conclusion section. Thank you for this valuable suggestion.

---

> > ### Author Response · Authors · 2025-11-22
> > **Response to Reviewer Avyq (2/3)**
> >
> > > **W4 & Q3: (1)** The current choice of baselines (PCMCI+, Bagged-PCMCI+, and NTS-NOTEARS) does not sufficiently represent the state of the art in time-series causal discovery. **(2)** At least one Granger-based causal discovery baseline, such as Neural Granger Causality (Tank et al., 2021 ) or SPACETIME (Mameche et al., 2025) should have been included for completeness. How does ECD-aMCI compare to nonlinear Granger models (e.g., NCG or RNN-based GC) in detecting lagged vs. contemporaneous dependencies?
> >
> > **R4:**  **(1)** In fact, we selected our baselines from a pool of ten algorithms covering the four major families of causal discovery methods (constraint-based, score-based, structural causal model-based, and Granger causality-based algorithms). We filtered these methods based on whether they learn a window causal graph and whether they handle contemporaneous effects, which resulted in five candidates (PCMCI+, Bagged-PCMCI+, NTS-NOTEARS, DYNOTEARS, and VARLINGAM). Following prior literature, we then selected the three state-of-the-art methods among them as our final baselines. As detailed in Appendix A.2 and Table 4, we have already explained the rationale behind choosing these three algorithms.
> >
> > **(2)**
> >
> > * Regarding Neural Granger Causality [2], the algorithm learns only a summary causal graph (not containing temporal lag information) and therefore is not directly comparable to the proposed algorithm or to the other selected baselines, which all aim to learn window causal graphs  (containing temporal lag information). To address the your concern, we additionally compressed the window causal graph produced by our method into a summary graph and conducted a separate comparison with Neural Granger Causality. The corresponding simulation results are provided below.
> >
> >   **Table 1:  F1-score$_{\text{all}}$ Comparison ($a$​ variation)**
> >
> >   |             | a=0.9           | a=0.8           | a=0.7           | a=0.6           | a=0.5           |
> >   | ----------- | --------------- | --------------- | --------------- | --------------- | --------------- |
> >   | ECD-aMCI    | 0.8899 ± 0.0643 | 0.9262 ± 0.0502 | 0.9343 ± 0.0474 | 0.9330 ± 0.0524 | 0.9269 ± 0.0544 |
> >   | Granger-RNN | 0.5642 ± 0.1193 | 0.6098 ± 0.0661 | 0.5986 ± 0.0609 | 0.5824 ± 0.0566 | 0.5639 ± 0.0558 |
> >
> > * Although the SPACETIME paper provides a link to the code (https://eda.group/spacetime), the core implementation file `spacetime.py` is not actually included, making it difficult to reproduce their results in a short time. Nevertheless, we will discuss the contributions of SPACETIME in the related work section and include its baseline methods (DYNOTEARS, VAR-LiNGAM) in our evaluation. The corresponding simulation results are provided below.
> >
> >   **Table 2:  F1-score$_{\text{all}}$ Comparison ($a$ variation)**
> >
> >   | Method        | a=0.9         | a=0.8         | a=0.7         | a=0.6         | a=0.5         |
> >   | ------------- | ------------- | ------------- | ------------- | ------------- | ------------- |
> >   | ECD-aMCI      | **0.87±0.05** | **0.92±0.04** | **0.93±0.04** | **0.93±0.03** | **0.93±0.03** |
> >   | PCMCI+        | 0.78±0.07     | 0.86±0.05     | 0.90±0.04     | 0.92±0.03     | 0.92±0.03     |
> >   | Bagged-PCMCI+ | 0.72±0.07     | 0.78±0.05     | 0.81±0.05     | 0.81±0.05     | 0.80±0.05     |
> >   | NTS-NOTEARS   | 0.35±0.12     | 0.56±0.16     | 0.75±0.06     | 0.85±0.04     | 0.88±0.04     |
> >   | VAR-LiNGAM    | 0.52±0.17     | 0.74±0.15     | 0.86±0.05     | 0.89±0.04     | 0.89±0.04     |
> >   | DYNOTEARS     | 0.27±0.07     | 0.23±0.08     | 0.17±0.08     | 0.12±0.07     | 0.08±0.06     |
> >
> > **Table 3: F1-lagged Comparison ($a$ variation)**
> > |Method|a=0.9|a=0.8|a=0.7|a=0.6|a=0.5|
> > |---|---|---|---|---|---|
> > |ECD-aMCI|**0.78±0.11**|**0.85±0.08**|**0.87±0.07**|**0.88±0.07**|**0.89±0.06**|
> > |PCMCI+|0.52±0.16|0.69±0.12|0.81±0.09|0.85±0.07|0.87±0.07|
> > |Bagged-PCMCI+|0.46±0.15|0.60±0.12|0.67±0.10|0.68±0.10|0.67±0.09|
> > |NTS-NOTEARS|0.24±0.15|0.49±0.19|0.71±0.09|0.80±0.07|0.84±0.07|
> > |VAR-LiNGAM|0.03±0.06|0.04±0.06|0.03±0.06|0.03±0.06|0.03±0.06|
> > |DYNOTEARS|0.29±0.20|0.60±0.21|0.77±0.08|0.81±0.07|0.81±0.07|
> >
> >  **Table 4: SHD Comparison ($a$ variation)**
> > |Method|a=0.9|a=0.8|a=0.7|a=0.6|a=0.5|
> > |---|---|---|---|---|---|
> > |ECD-aMCI|**11.60±4.10**|**8.90±3.17**|**8.64±3.11**|**9.41±3.12**|10.55±3.13|
> > |PCMCI+|16.67±4.56|12.57±3.39|10.56±3.04|10.54±2.96|11.21±3.00|
> > |Bagged-PCMCI+|22.36±5.08|18.81±3.89|18.01±4.01|19.24±4.46|20.48±4.33|
> > |NTS-NOTEARS|67.85±16.69|40.21±11.27|23.32±5.58|14.71±4.07|11.69±3.59|
> > |VAR-LiNGAM|41.62±2.42|42.57±2.48|43.53±2.23|44.33±1.89|45.14±1.99|
> > |DYNOTEARS|37.64±11.90|22.17±12.17|12.57±4.44|10.47±3.63|**10.37±3.53**|

---

> > > ### Author Response · Authors · 2025-11-22
> > > **Response to Reviewer Avyq (3/3)**
> > >
> > > > **W5:** There is no formal quantification of how much adaptive conditioning improves causal recovery beyond qualitative reasoning about signal-to-noise ratios.
> > >
> > > **R5:** Our justification relies on reasoning about signal-to-noise ratios, which illustrate why the proposed method enhances the effectiveness of CI testing. The exact magnitude of improvement, however, depends on the specific data-generating conditions and is therefore demonstrated through extensive simulation studies under a variety of settings.
> > >
> > >
> > >
> > > > **Q1:** Could a scaled-down Bagged-PCMCI+ comparison be included for nonlinear/fMRI data to complete the baseline picture?
> > >
> > > **R6:** We will include the scaled-down Bagged-PCMCI+ comparison for nonlinear/fMRI data in the revised manuscript.
> > >
> > >
> > >
> > > **Reference**
> > >
> > > [1] Mameche, S., Cornanguer, L., Ninad, U., & Vreeken, J. (2025, April). SPACETIME: Causal Discovery from Non-Stationary Time Series. In *Proceedings of the AAAI Conference on Artificial Intelligence* (Vol. 39, No. 18, pp. 19405-19413).
> > >
> > > [2] Tank, A., Covert, I., Foti, N., Shojaie, A., & Fox, E. B. (2021). Neural granger causality. *IEEE Transactions on Pattern Analysis and Machine Intelligence*, *44*(8), 4267-4279.

---

> > > > ### Comment · Reviewer_Avyq · 2025-11-24
> > > >
> > > > I appreciate the authors' effort in addressing the review comments and questions. My apologies for the incorrect reference to the experimental section in the appendix. I meant to say A.5, not C.3. I revised my score (from 4 to 6).

---

> > > > > ### Author Response · Authors · 2025-11-24
> > > > >
> > > > > We sincerely thank you for the comment and the updated evaluation. We are glad that our clarifications helped resolve the concerns.

---

### Note · Authors · 2025-12-08

I have read and agree with the venue's withdrawal policy on behalf of myself and my co-authors.